# EndoBench: A Comprehensive Evaluation of Multi-Modal Large Language Models for Endoscopy Analysis

**Shengyuan Liu**[1][*]    **Boyun Zheng**[1][*]    **Wenting Chen**[2][*]    **Zhihao Peng**[1]

**Zhenfei Yin**[3]    **Jing Shao**[4]    **Jiancong Hu**[5]    **Yixuan Yuan**[1][†]

[1]Chinese University of Hong Kong    [2]City University of Hong Kong
[3]University of Oxford    [4] Shanghai AI Laboratory
[5] The Sixth Affiliated Hospital, Sun Yat-sen University

## Abstract

Endoscopic procedures are essential for diagnosing and treating internal diseases, and multi-modal large language models (MLLMs) are increasingly applied to assist in endoscopy analysis. However, current benchmarks are limited, as they typically cover specific endoscopic scenarios and a small set of clinical tasks, failing to capture the real-world diversity of endoscopic scenarios and the full range of skills needed in clinical workflows. To address these issues, we introduce EndoBench, the first comprehensive benchmark specifically designed to assess MLLMs across the full spectrum of endoscopic practice with multi-dimensional capacities. EndoBench encompasses 4 distinct endoscopic scenarios, 12 specialized clinical tasks with 12 secondary subtasks, and 5 levels of visual prompting granularities, resulting in 6,832 rigorously validated VQA pairs from 21 diverse datasets. Our multi-dimensional evaluation framework mirrors the clinical workflow—spanning anatomical recognition, lesion analysis, spatial localization, and surgical operations—to holistically gauge the perceptual and diagnostic abilities of MLLMs in realistic scenarios. We benchmark 23 state-of-the-art models, including general-purpose, medical-specialized, and proprietary MLLMs, and establish human clinician performance as a reference standard. Our extensive experiments reveal: (1) proprietary MLLMs outperform open-source and medical-specialized models overall, but still trail human experts; (2) medical-domain supervised fine-tuning substantially boosts task-specific accuracy; and (3) model performance remains sensitive to prompt format and clinical task complexity. EndoBench establishes a new standard for evaluating and advancing MLLMs in endoscopy, highlighting both progress and persistent gaps between current models and expert clinical reasoning. We publicly release our benchmark and code.

## 1 Introduction

Gastrointestinal and urinary system diseases pose significant global health challenges, where early detection is critical, especially for cancers often diagnosed at advanced stages [1, 2, 3, 4]. Endoscopy is a vital diagnostic and therapeutic tool, enabling visualization of internal organs across medical specialties [5, 6, 7]. As the gold standard for examining internal structures, endoscopy aids in timely

---

[*]Equal contributions.

[†]Corresponding author (yxyuan@ee.cuhk.edu.hk).

39th Conference on Neural Information Processing Systems (NeurIPS 2025) Track on Datasets and Benchmarks.

Table 1: Comparisons with existing multi-modal endoscopic benchmarks.

| Benchmark | Size | Scenario | Task | Granularity | Data Source |
|---|---|---|---|---|---|
| *OmniMedVQA [21] | 1877 | - | 3 | 1 | 3 Public |
| *GMAI-MMBench [22] | 3749 | - | 7 | 4 | 16 Public |
| Kvasir-VQA [23] | 6500 | GS,CS | 6 | 1 | 2 Public |
| Surgical-VQA [24] | 54K | SG | 5 | 1 | 2 Public |
| SSG-VQA [25] | 960K | SG | 5 | 2 | 3 Public |
| EndoChat [17] | 396K | SG | 5 | 2 | 3 Public |
| ColonINST [13] | 300K+ | CS | 4 | 1 | 19 Public |
| **EndoVQA-Instruct** | 446K+ | GS,CS,CE,SE | 12 | 5 | 20 Public, 1 In-House |
| **EndoBench** | 6832 | GS,CS,CE,SE | 12 | 5 | 20 public, 1 In-House |

Abbreviation: GS for Gastroscopy, CS for Colonoscopy, CE for Capsule endoscopy, SE for Surgical endoscopy.
* The endoscopic data of this benchmark.

pathology detection [8, 9]. However, the rising demand for endoscopic procedures highlights the need for advanced technologies like artificial intelligence to improve accuracy and efficiency [10, 11, 12].

Recent advances in Multi-modal Large Language Models (MLLMs) have produced numerous specialized medical MLLMs dedicated to endoscopy analysis [13, 14, 15, 16, 17, 18, 19, 20]. These models enable users to interact through text prompts when analyzing endoscopic images, facilitating various clinical tasks including surgical instrument identification [15, 16], lesion detection [13], endoscopic image caption [13, 17], and so on. As these endoscopy-focused MLLMs have developed, there has been a parallel need for robust evaluation frameworks to assess their clinical utility and performance. Existing benchmarks can be categorized into general-purpose [21, 22] and endoscopy-specific evaluations [23, 24, 25, 13]. General benchmarks provide comprehensive assessments across diverse medical data but typically include only limited endoscopic samples covering a narrow range of tasks. Endoscopy-specific benchmarks [23, 24, 25, 13] focus on common procedures such as surgical and colonoscopy, evaluating performance on procedure-specific tasks. Despite these efforts, current benchmarks face significant challenges in assessing whether MLLMs can truly comprehend gastrointestinal endoscopic scenarios with the depth and nuance of clinical professionals.

The first challenge lies in the limited scope of existing endoscopic benchmarks, which typically focus on specific scenarios. For instance, Surgical-VQA [24] and SSG-VQA [25] primarily evaluate the surgical scenario, while ColonINST [13] concentrates exclusively on colonoscopy. In real clinical settings, however, clinicians always navigate across diverse endoscopic scenarios including Capsule endoscopy (CE), Gastroscopy (GS), Colonoscopy (CS), and Surgical endoscopy (SE). These scenarios differ substantially in their imaging characteristics, anatomical focus, and clinical objectives—ranging from diagnostic screening to interventional procedures—requiring clinicians to possess versatile expertise. The compartmentalized nature of current benchmarks fails to comprehensively assess whether MLLMs can adapt to this multi-modal reality of endoscopic practice. A more holistic, ***multi-scenario evaluation*** framework that spans all endoscopy scenarios is therefore essential to accurately gauge the clinical utility of these models.

Another challenge is that existing endoscopic VQA benchmarks [23, 24, 25, 13] evaluate only a limited range of tasks, ignoring the multi-dimensional capacities required in clinical practice. While benchmarks like Kvasir-VQA [23] focus on basic recognition tasks and ColonINST [13] emphasizes lesion classification, actual clinical endoscopy follows a structured workflow requiring progressively more sophisticated analysis [26, 27]. Clinicians always identify organs, recognize anatomical landmarks, detect and classify lesions, quantify findings, precisely localize abnormalities, perform pre-surgical assessments, and execute appropriate interventions. This clinical process demands capabilities spanning from whole-image interpretation to detailed region-level analysis. However, current benchmarks, by focusing on limited tasks rather than this comprehensive spectrum of abilities, inadequately evaluate whether MLLMs can replicate the nuanced expertise that characterizes expert endoscopic assessment. Therefore, a more holistic evaluation framework is needed that assesses model performance across the ***multi-dimensional capacities*** required in clinical endoscopic examinations.

To address these challenges, we introduce **EndoBench**, a comprehensive endoscopy benchmark designed to evaluate the multi-dimensional capabilities of current multi-modal large language models

(MLLMs) in endoscopic image analysis in Fig. 2. To the best of our knowledge, EndoBench is the most extensive multi-modal endoscopic benchmark to date, encompassing 4 distinct endoscopic scenarios, 12 specialized endoscopic tasks with 12 secondary subtasks, and 5 levels of visual prompting granularities, as detailed in Table 1. For *multi-scenario coverage*, EndoBench spans the complete spectrum of endoscopy procedures—from Gastroscopy and Colonoscopy to Capsule endoscopy and Surgical endoscopy. For *multi-dimensional capacities evaluation*, EndoBench assesses MLLMs from 12 distinct tasks across 4 major categories, including anatomical structure recognition, lesion analysis and grading, spatial localization and region understanding, and surgical workflow and operation analysis. To thoroughly evaluate fine-grained perceptual capabilities, we implement 5 visual prompting granularities—image-level, box-level, contour-level, multi-box, and multi-contour. Our dataset construction involves collecting 20 public and 1 private endoscopy datasets and standardizing QA pairs, yielding 446,535 VQA pairs comprising our EndoVQA-Instruct dataset, the current largest endoscopic instruction-tuning collection. From EndoVQA-Instruct, we extract representative pairs that undergo rigorous clinical review, resulting in our final EndoBench dataset of 6,832 clinically validated VQA pairs. For rigorous evaluation, we evaluate 13 open-source general-purpose MLLMs, 5 specialized medical MLLMs, and 5 proprietary general-purpose MLLMs on EndoBench. To establish clinical reference standards, we recruit two certified clinicians to answer questions from EndoBench. Extensive experiments show that while proprietary MLLMs outperform open-source and specialized models overall, they still lag behind human experts.

The main contributions of this paper are summarized as follows:

- We introduce EndoBench, the first comprehensive benchmark specifically designed to evaluate MLLMs across the complete spectrum of endoscopy, covering 4 endoscopic scenarios, 12 specialized tasks with 12 secondary subtasks, and 5 levels of visual prompting granularities.

- We develop the multi-dimensional evaluation framework that mirrors the clinical workflow progression from basic anatomical recognition to advanced surgical intervention, assessing MLLMs' capabilities across the full spectrum of endoscopic analysis skills.

- We conduct the extensive comparative evaluation of 23 MLLMs (13 open-source general-purpose, 5 medical-specialized, and 5 proprietary models) against human clinician performance, providing insights into current model capabilities.

## 2 Related Work

### 2.1 Multi-Modal Large Language Models (MLLMs)

Multi-Modal Large Language Models (MLLMs) are adept at addressing complex multi-modal tasks through large-scale pretraining. Early models like BLIP [28, 29, 30] and Flamingo [31] used joint encoders with cross-attention for processing images and text. Later, auto-regressive MLLMs emerged, tokenizing images into visual tokens combined with text tokens for LLM input. Instruction-tuned models like LLaVA [32] achieved strong results on vision-language tasks. Recent MLLMs [33, 34, 35, 36, 37, 38, 39, 40], including QwenVL [41] and InternVL [38], scaled these architectures, rivaling commercial models like GPT-4o [42] and Claude-3.7 [43]. In the medical field, recent studies [44, 45, 46, 47, 48, 49, 50, 51, 52, 53, 54] have focused on fine-tuning general-purpose models on medical datasets. LLaVA-Med [44] enhanced LLaVA using PMC-15M [55] for improved medical VQA performance. HuatuoGPT-Vision [47] created 1.3M medical VQA samples from PubMed, while MedDr [48] employed a retrieval-based approach using InternVL [38]. Other works [13, 56, 24, 15, 57] explored MLLMs in endoscopy. ColonGPT [13] aids endoscopists with dialogues, while SurgicalGPT [14] and Surgical-LVLM [56] demonstrated surgical scenario understanding.

### 2.2 Benchmark for Medical MLLMs

In the rapidly advancing field of medical MLLMs [58], numerous benchmarks [21, 59, 60, 22, 47, 61, 62, 63, 64, 65, 66, 67] have been developed, offering large and diverse medical VQA datasets to enhance evaluation robustness. GMAI-MMBench [22] incorporated 284 datasets across 38 imaging modalities, 18 clinical tasks, and 18 medical departments. Medifusion [60] introduced a benchmark with confusing image pairs, requiring distinct answers for identical questions based on subtle image differences. Despite these advancements, endoscopic data remains underrepresented.

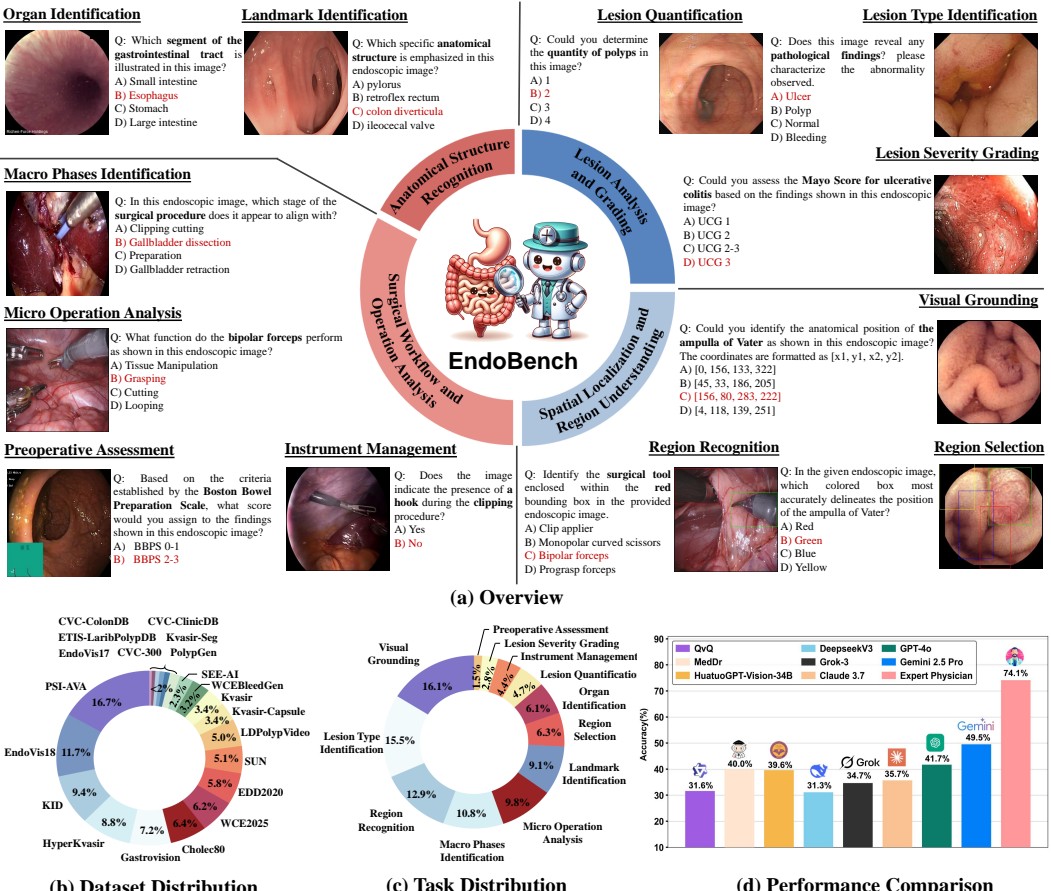

Figure 1: Overview of our **EndoBench**, the first comprehensive benchmark specifically designed to evaluate MLLMs across the complete spectrum of endoscopy, covering 4 endoscopic scenarios, 12 specialized tasks with 12 secondary subtasks, and 5 levels of visual prompting granularities.

In the endoscopic domain, most benchmarks [24, 23, 25, 13, 68] focus on specific applications. SurgicalVQA [24] used Cholec80 [69] and EndoVis-18 [70] to evaluate vision-language models in surgery. SSG-VQA [25] tackled laparoscopic tasks like geometric localization and procedure analysis. Kvasir-VQA [23] added 6,500 question-answer pairs to HyperKvasir [71] and Kvasir-Instrument [72]. ColonINST [13] targeted colonoscopy with 303,001 images from 19 datasets. However, these benchmarks lack scenario diversity and task scope, failing to reflect real clinical scenarios. Thus, we propose EndoBench, a comprehensive benchmark for evaluating MLLMs across diverse endoscopy applications.

# 3 EndoBench

**Overview.** EndoBench is a comprehensive MLLM evaluation framework spanning 4 endoscopy scenarios and 12 clinical tasks with 12 secondary subtasks that mirror the progression of endoscopic examination workflows. Featuring 5 levels of visual prompting granularities to assess region-specific understanding, our EndoBench contains 6,832 clinically validated VQA pairs derived from 21 endoscopy datasets. This structure enables precise measurement of MLLMs' clinical perceptual, diagnostic accuracy, and spatial comprehension across diverse endoscopic scenarios.

## 3.1 Benchmark Construction

This section introduces the three main construction steps of EndoBench, as shown in Fig. 2.

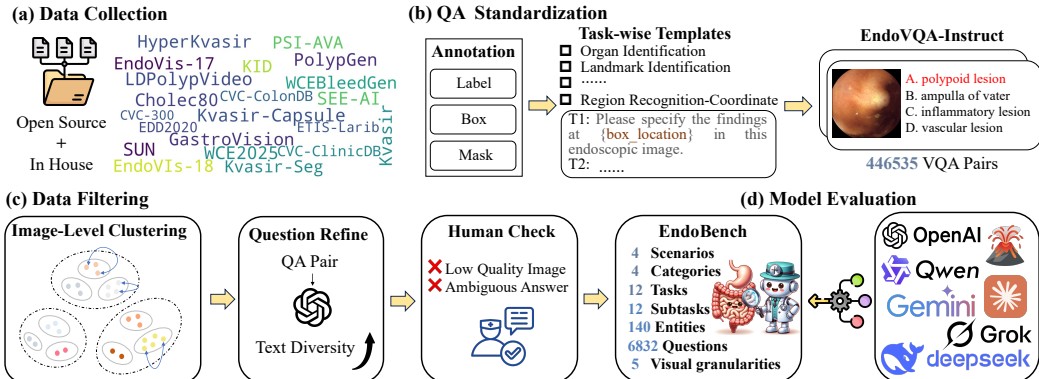

Figure 2: Data construction process of **EndoBench**, consisting of (a) data collection, (b) QA standardization, and (c) data filtering. Finally, we implement (d) model evaluation on EndoBench.

**Data Collection.** The foundation of our benchmark lies in thorough data collection. Endoscopic images can be classified into two primary categories based on their clinical applications. The first category encompasses diagnostic images used for observation and documentation, including routine upper gastrointestinal endoscopy, lower gastrointestinal colonoscopy, and capsule endoscopy images. The second category consists of therapeutic images used in image-guided minimally invasive procedures, specifically endoscopic surgical images. To ensure comprehensive coverage, on the one hand, we gather 20 public endoscopy datasets from online sources to encompass various endoscopic image types and professional terminology, including Kvasir [73], HyperKvasir [71], Kvasir-Capsule [74], GastroVision [75], KID [76], WCEBleedGen [77], SEE-AI [78], Kvasir-Seg [79], CVC-ColonDB [80], ETIS-Larib [81], CVC-ClinicDB [82], CVC-300 [83], EDD2020 [84], SUN-Database [85], LDPolypVideo [86], PolypGen [87], Cholec80 [69], EndoVis-17 [88], EndoVis-18 [70], and PSI-AVA [68]. On the other hand, we further enhance the data diversity by incorporating a private wireless capsule endoscopy image dataset from partner hospitals. All data undergoes the privacy de-identification in accordance with medical ethics requirements. Detailed dataset information is provided in Appendix.

**QA Standardization.** Following data collection, we standardize the diverse annotation formats across datasets, which include anatomical landmark and pathological lesion labels, as well as structural or lesion annotations in coordinate or image formats (bounding boxes and masks). In collaboration with professional physicians, we develop 12 specialized tasks across 4 major categories and map each dataset's attributes to the appropriate tasks. The 12 specialized tasks include with 12 secondary subtasks. We create 5-8 distinct question templates per task. To facilitate evaluation, each QA pair is supplemented with incorrect answer options, transforming it into a multiple-choice format. Entity labels are naturally conducive to QA pair construction. Question templates are designed based on original categories, with incorrect options randomly selected from attribute nouns of the same type. For spatial comprehension tasks, we standardize coordinate formatting as $[x1, y1, x2, y2]$. To generate plausible distractors for the visual prompts, we create alternative boxes with dimensions similar to the actual bounding box while maintaining overlap below specified thresholds. This process yields 446,535 image-text pairs, comprising our EndoVQA-Instruct dataset.

**Data Filtering.** To create a balanced and representative subset for MLLM evaluation, we implement a systematic filtering pipeline on the EndoVQA-Instruct dataset to obtain our EndoBench. We first balance the entity distribution, then employ the DINO-V2 [89] vision foundational model to extract visual embeddings within entities, capturing fine-grained latent representation. Using K-center clustering, we select representative images within each class while maintaining categorical balance and eliminating noise. To enhance dataset diversity and comprehensively evaluate MLLM capabilities, we utilize the GPT-4o-mini API to reformulate questions from the original QA pairs, varying expression styles and syntactic structures while preserving semantic content. This approach enables assessment of MLLMs' adaptability to diverse linguistic representations. Finally, two professional physicians conduct a thorough review of questions and answers, eliminating substandard images and incorrect or ambiguous responses to ensure data quality. Finally, these processes result in our final EndoBench dataset, including 6,832 VQA pairs.

## 3.2 Multi-Scenarios Coverage

Although clinicians regularly navigate diverse endoscopy scenarios in real clinical practice, existing endoscopic benchmarks [23, 24, 25, 13] focus on limited scenarios, making evaluations insufficiently comprehensive. To address this gap, EndoBench encompasses all four major types of endoscopy: Gastroscopy (GS), Colonoscopy (CS), Capsule Endoscopy (CE), and Surgical Endoscopy (SE). The dataset includes 583 samples of GS (8.53%), 1,848 samples of CS (27.05%), 1,678 samples of CE (24.56%), and 2,723 samples of SE (39.86%), totaling 6,832 samples across all endoscopy types.

## 3.3 Multi-Dimensional Capacity

Although clinical endoscopy follows a structured workflow requiring progressively more sophisticated analysis [26, 27], existing endoscopy benchmarks [23, 24, 25, 13] evaluate only a limited range of tasks, overlooking the multi-dimensional capacities required in clinical practice. To comprehensively evaluate these capacities in MLLMs, EndoBench encompasses 12 clinical tasks with 12 secondary subtasks across 4 major categories for endoscopy analysis. These categories include: (1) *anatomical structure recognition* (organ identification, landmark identification); (2) *lesion analysis and grading* (lesion quantification, lesion type identification, lesion severity grading); (3) *spatial localization and region understanding* (visual grounding, region selection, region recognition); and (4) *surgical workflow and operation analysis* (preoperative assessment, macro phases identification, micro operation analysis, instrument management). The secondary subtasks are detailed in Appendix. The distribution of these tasks is illustrated in Fig. 2 (b). Moreover, to thoroughly evaluate fine-grained perceptual capabilities of current MLLMs, we implement 5 levels of visual prompting granularities, including image-level, box-level, contour-level, multi-box, and multi-contour. The contour and box are obtained from the original segmentation mask.

# 4 Experiments and Analysis

## 4.1 Experiment Setup

**Model Evaluation.** We evaluate 23 MLLMs, comprising 13 open-source models, 5 proprietary models, and 5 medical-domain-specific models. The open-source models, ranging from 3B to 72B parameters, include LLaVA [32], LLaVA-Next [90], CogVLM [33], ShareGPT-4v [34], Qwen2.5-VL [41], Janus-Pro [36], InternVL2.5 [38], and QvQ [91]. Among the proprietary models, we evaluate three reasoning-focused models (Deepseek-V3 with vision, Claude-3.7-Sonnet [43], and Gemini-2.5-Pro) and two other MLLMs (GPT-4o [92] and Grok-3[93]). For medical models, we assess MedDr [48], LLaVA-Med [44], HuatuoGPT-Vision [47], and the endoscopy-specialized ColonGPT [13].

**Human Study.** To establish a benchmark for performance, the study includes an evaluation of human clinicians. We randomly select 255 questions from our EndoBench across all the sub-tasks except coordinate-related tasks, due to the precise coordinate format being unsuitable for intuitive human judgment. Each sub-task may include 5, 10, or 15 samples. Two certified clinicians with expertise in endoscopy independently finished the selected questions. Their scores are averaged for each task to provide a reference standard for comparison.

**Evaluation Metrics.** To evaluate model performance, we measure the accuracy by counting exact matches between predictions and ground-truth answers. For some medical-focused MLLMs that struggle with formatting responses, we use Qwen2.5-VL-72B [41] to extract plausible answers for matching. If no valid answer is found, the sample is marked as an error.

## 4.2 Experimental Results

**Results across Capacities.** Table 2 summarizes the performance of various MLLMs across 12 clinical tasks. Proprietary models dominate, with Gemini-2.5-Pro achieving the highest average score (49.53%) and excelling in 7 tasks, though still far from human physician performance (74.12%). GPT-4o follows with strong results (41.69%), especially in lesion analysis. Medical-domain models like MedDr-80B and HuatuoGPT-Vision-34B perform well in specific tasks but lag overall. ColonGPT shows extreme variability, excelling in Preoperative Assessment (95%, surpassing physicians at 80%) but underperforming elsewhere. Open-source models generally trail, though QvQ-72B (31.62%)

Table 2: Results of different MLLMs on 12 clinical tasks in EndoBench. The best-performing model in each category is **in-bold**, and the second best is underlined.

| MLLMs | Avg | Anatomy | | Lesion | | | Surgery | | | | Spatial | | |
|---|---|---|---|---|---|---|---|---|---|---|---|---|---|
| | | LI | OI | LQ | LT | LS | PA | MP | MO | IM | VG | RS | RR |
| Random | 24.96 | 23.68 | 24.64 | 23.25 | 15.98 | 15.48 | 50.00 | 25.61 | 27.78 | 33.33 | 28.10 | 22.63 | 24.26 |
| Physician | 74.12 | 93.33 | 65.00 | 70.00 | 66.67 | 46.67 | 80.00 | 60.00 | 77.14 | 80.00 | - | 93.33 | 80.00 |
| **Open-Source MLLMs** | | | | | | | | | | | | | |
| Llava-v1.5-7B | 26.62 | 22.24 | 25.84 | 21.64 | 22.16 | 15.79 | 52.00 | 24.25 | 40.24 | 35.69 | 22.87 | 25.64 | 28.99 |
| Llava-v1.5-13B | 24.99 | 24.80 | 19.14 | 19.75 | 12.89 | 18.89 | 56.00 | 22.36 | 46.55 | 34.68 | 19.73 | 26.56 | 22.58 |
| Llava-llama3-8B | 24.75 | 22.56 | 21.77 | 22.40 | 18.56 | 18.27 | 51.00 | 23.71 | 42.49 | 37.04 | 18.68 | 25.40 | 20.82 |
| Llava-Next-Llama3-8B | 25.10 | 27.04 | 23.21 | 17.01 | 18.04 | 17.03 | 51.00 | 24.53 | 44.89 | 39.73 | 18.24 | 26.79 | 22.27 |
| CogVLM-Chat-7B | 27.58 | 24.64 | 25.60 | 21.08 | 20.10 | 18.58 | 50.00 | 23.71 | 41.44 | 34.68 | 26.76 | 24.94 | 31.27 |
| ShareGPT-4v | 18.35 | 11.52 | 18.42 | 19.19 | 1.55 | 14.24 | 0.00 | 16.26 | 31.38 | 21.55 | 11.51 | 12.70 | 25.02 |
| Qwen2.5VL-3B-Instruct | 25.01 | 21.60 | 23.44 | 20.70 | 15.98 | 16.72 | 44.00 | 26.02 | 42.34 | 50.17 | 17.49 | 25.87 | 21.05 |
| Qwen2.5VL-7B-Instruct | 27.63 | 22.24 | 25.84 | 21.64 | 22.16 | 15.79 | 55.00 | 24.25 | 40.24 | 57.91 | 22.87 | 25.64 | 28.99 |
| Qwen2.5VL-72B-Instruct | 27.25 | 28.48 | 20.10 | 22.21 | 12.37 | 14.55 | 53.00 | 27.51 | 48.05 | 50.17 | 22.87 | 26.10 | 23.11 |
| Janus-Pro-7B | 28.81 | 25.28 | 23.68 | 22.31 | 13.40 | 17.03 | 50.00 | 27.37 | 47.90 | 45.45 | 25.56 | 25.87 | 30.89 |
| InternVL2.5-8B | 27.96 | 23.20 | 20.10 | 19.09 | 8.76 | 17.96 | 54.00 | 26.83 | 49.25 | 45.79 | 16.74 | 26.10 | 35.32 |
| InternVL2.5-38B | 30.09 | 28.48 | 31.82 | 22.02 | 13.40 | 19.50 | 57.00 | 27.64 | 47.60 | 49.83 | 19.13 | 28.18 | 34.10 |
| QvQ-72B | 31.62 | 22.08 | 15.31 | 30.91 | 22.68 | 18.89 | 53.00 | 28.86 | 49.85 | 53.87 | 28.85 | 37.88 | 31.35 |
| **Open-Source Medical-Domain MLLMs** | | | | | | | | | | | | | |
| MedDr-80B | 39.96 | **56.00** | **43.06** | 36.96 | 21.65 | 17.65 | 52.00 | 28.05 | 57.51 | 48.48 | 45.14 | 47.58 | 31.73 |
| Llava-Med-7B | 24.71 | 41.44 | 26.79 | 15.79 | 24.23 | 8.67 | 47.00 | 17.61 | 24.93 | 25.26 | 24.36 | 37.88 | 25.17 |
| HuatuoGPT-Vision-7B | 35.57 | 34.88 | 39.47 | 37.43 | 23.20 | 21.05 | 45.00 | 31.03 | 49.85 | 49.49 | 24.07 | 46.42 | 32.26 |
| HuatuoGPT-Vision-34B | 39.58 | 36.64 | 33.25 | 34.97 | 19.59 | 21.98 | 86.00 | 35.77 | 55.26 | 59.26 | 31.69 | 60.51 | 37.30 |
| ColonGPT | 15.60 | 30.40 | 11.00 | 27.69 | 12.37 | 0.00 | **95.00** | 5.42 | 1.65 | 15.83 | 2.99 | 4.62 | 21.36 |
| **Proprietary MLLMs** | | | | | | | | | | | | | |
| Deepseek-V3 | 31.13 | 24.96 | 21.53 | 29.96 | 21.65 | 20.43 | 46.00 | 27.91 | 52.10 | 54.55 | 21.38 | 38.57 | 29.37 |
| Grok-3 | 34.66 | 32.16 | 26.08 | 33.65 | 16.49 | 22.29 | 56.00 | 27.38 | 49.25 | 53.87 | 20.93 | 54.73 | 36.23 |
| Claude-3.7-Sonnet | 35.67 | 34.56 | 26.07 | 36.01 | 29.90 | **25.39** | 44.00 | 27.78 | 53.76 | 48.48 | 27.65 | 51.27 | 33.03 |
| GPT-4o | 41.69 | 44.16 | 33.73 | **42.25** | **39.69** | 24.15 | 92.00 | 41.19 | 59.16 | **63.63** | 27.06 | 41.80 | 37.22 |
| Gemini-2.5-Pro | **49.53** | 44.16 | 39.71 | 41.97 | 29.38 | 24.46 | 90.00 | **46.21** | **67.87** | 62.96 | **50.52** | **73.21** | **48.59** |

Abbreviation: Anatomy for Anatomical Structure Recognition, Lesion for Lesion Analysis and Grading, Surgery for Surgical Workflow and Operation Analysis, Spatial for Spatial localization and region understanding. The abbreviations for the corresponding tasks are defined in Appendix.

and InternVL2.5-38B (30.09%) show promise. These findings reveal proprietary MLLMs' overall superiority but underscore the gap between MLLMs and human expertise. Fig. 3 illustrates the performance of MLLMs across four major categories. Most models consistently achieve their highest performance on surgery-related tasks, while struggling most with lesion-relevant tasks, highlighting the varying difficulty levels these categories present for current MLLMs.

**Results across Scenarios.** Table 3 compares performance of different MLLMs across endoscopy scenarios. Proprietary MLLMs, particularly Gemini-2.5-Pro (52.39%), outperform all other models across clinical tasks and visual prompts, with GPT-4o (42.87%) following as a strong competitor. Medical-domain models, like HuatuoGPT-Vision-34B (41.55%), show potential in specific tasks but lack consistency. Open-source models generally underperform, though QvQ-72B (33.01%) and InternVL2.5-38B (32.36%) demonstrate some promise. ColonGPT is a notable outlier, excelling in Preoperative Assessment (95%) but performing poorly overall (10.47%). Despite advancements, all MLLMs lag human physicians, who achieve a superior average score of 76.64%, emphasizing the need for further development to bridge this performance gap.

## 4.3 Discussion

From the above results, four key insights have been deduced as follows:

**1) Endoscopy remains a challenging domain for MLLMs, with significant gaps between models and human expertise.** Human experts achieve an average accuracy of 74.12% in endoscopy tasks, while the top-performing model, Gemini-2.5-Pro, reaches only 49.53%—a gap of roughly 25%. This highlights the inherent difficulty of endoscopy, which demands both precise visual interpretation and specialized medical knowledge. Proprietary models consistently outperform open-source models

Table 3: Results of different MLLMs on 4 different endoscopy scenarios and 4 different visual prompts in EndoBench. The best-performing model in each category is **in-bold**, and the second best is underlined.

| MLLMs | Endoscopy Scenarios | | | | | Viusal Prompt | | | | |
|---|---|---|---|---|---|---|---|---|---|---|
| | Avg | GS | CS | CE | SE | Avg | Box | Cont | Mul | Coor |
| Random | 25.58 | 24.01 | 23.41 | 25.48 | 26.35 | 23.61 | 26.10 | 19.23 | 25.10 | 24.02 |
| Physician | 76.64 | 80.00 | 81.94 | 78.26 | 74.59 | 80.00 | 86.67 | 73.33 | 80.00 | - |
| **Open-Source MLLMs** | | | | | | | | | | |
| Llava-v1.5-7B | 27.30 | 15.09 | 28.50 | 22.35 | 30.49 | 29.11 | 34.18 | 32.97 | 24.33 | 24.94 |
| Llava-v1.5-13B | 26.22 | 15.61 | 22.14 | 20.86 | 30.23 | 23.57 | 19.17 | 23.08 | 31.94 | 20.09 |
| Llava-llama3-8B | 26.87 | 27.79 | 19.86 | 22.47 | 28.96 | 20.74 | 19.17 | 17.58 | 24.71 | 21.48 |
| Llava-Next-Llama3-8B | 27.02 | 23.33 | 21.35 | 22.41 | 29.82 | 22.96 | 22.63 | 21.98 | 29.66 | 17.55 |
| CogVLM-Chat-7B | 29.23 | 18.52 | 26.32 | 23.18 | 33.17 | 30.96 | 31.87 | 29.67 | 30.42 | 31.87 |
| ShareGPT-4v-7B | 19.11 | 12.52 | 18.17 | 15.97 | 21.24 | 25.06 | 29.10 | 29.12 | 19.39 | 22.63 |
| Qwen2.5VL-3B-Instruct | 27.81 | 29.85 | 18.17 | 22.41 | 30.41 | 22.05 | 18.48 | 24.18 | 26.62 | 18.94 |
| Qwen2.5VL-7B-Instruct | 20.95 | 14.58 | 21.77 | 20.02 | 22.25 | 29.11 | 34.18 | 32.97 | 24.33 | 24.94 |
| Qwen2.5VL-72B-Instruct | 29.57 | 25.56 | 22.99 | 23.66 | 32.87 | 23.76 | 21.02 | 24.73 | 27.38 | 21.94 |
| Janus-Pro-7B | 31.12 | 26.93 | 23.99 | 24.20 | 34.77 | 30.83 | 32.33 | 26.92 | 36.12 | 27.94 |
| InternVL2.5-8B | 29.94 | 20.75 | 27.44 | 21.39 | 33.99 | 34.99 | 40.42 | 34.07 | 33.84 | 31.64 |
| InternVL2.5-38B | 32.36 | 28.99 | 25.85 | 26.64 | 35.48 | 33.48 | 38.57 | 31.32 | 31.94 | 32.10 |
| QvQ-72B | 33.01 | 31.73 | 29.93 | 25.03 | 35.48 | 30.88 | 34.41 | 31.32 | 26.62 | 31.18 |
| **Open-Source Medical-Domain MLLMs** | | | | | | | | | | |
| MedDr-80B | 40.92 | 51.46 | 37.76 | 38.50 | 39.92 | 31.73 | 33.03 | 34.62 | 27.38 | 31.87 |
| Llava-Med-7B | 25.11 | 35.33 | 24.10 | 23.06 | 23.67 | 24.71 | 25.64 | 23.08 | 23.57 | 26.56 |
| HuatuoGPT-Vision-7B | 36.04 | 36.88 | 34.32 | 35.22 | 36.38 | 32.40 | 32.56 | 35.71 | 28.52 | 32.79 |
| HuatuoGPT-Vision-34B | 41.55 | 45.80 | 38.14 | 33.61 | 42.97 | 37.20 | 39.49 | 37.91 | 35.36 | 36.03 |
| ColonGPT | 10.47 | 9.61 | 33.37 | 16.51 | 4.85 | 21.55 | 24.71 | 34.07 | 4.56 | 22.86 |
| **Proprietary MLLMs** | | | | | | | | | | |
| Deepseek-V3 | 32.34 | 27.79 | 30.46 | 27.53 | 34.59 | 29.86 | 31.18 | 34.07 | 27.38 | 26.79 |
| Grok-3 | 35.37 | 34.31 | 31.30 | 36.00 | 36.27 | 34.86 | 41.57 | 32.97 | 27.00 | 37.88 |
| Claude-3.7-Sonnet | 36.26 | 37.91 | 35.01 | 34.15 | 36.60 | 33.12 | 30.48 | 35.71 | 30.04 | 36.26 |
| GPT-4o | 42.87 | 45.97 | 43.54 | 34.86 | 43.72 | 36.78 | 32.79 | 35.71 | 34.98 | 43.65 |
| Gemini-2.5-Pro | **52.39** | **57.29** | **44.60** | **44.22** | **54.60** | **47.39** | **49.19** | **38.46** | **51.33** | **50.58** |

Abbreviation: Cont for Contour, Mul for Multi-region, Coor for Coordinate.

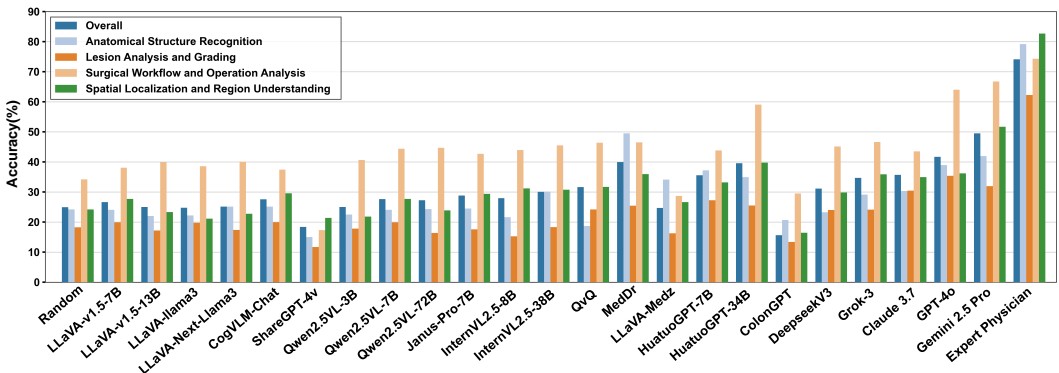

Figure 3: Performance comparison across 4 major categories in EndoBench among existing MLLMs.

overall, yet open-source models show a surprising edge in surgical scenarios, where their accuracy improves markedly compared to random baselines. In contrast, for non-surgical tasks like landmark and organ identification, open-source models perform no better than random guessing. This disparity suggests that while open-source models can leverage structured contexts, they falter in knowledge-intensive tasks, pointing to a need for enhanced domain-specific capabilities.

**2) Medical domain-specific Supervised Fine-Tuning markedly boosts model performance.** Medical models that underwent domain-specific supervised fine-tuning, such as MedDr and HuatuoGPT-Vision-34B, perform exceptionally well in tasks like landmark identification and organ recognition,



Figure 4: Case study reveals that model performance varies with different visual prompt formats.

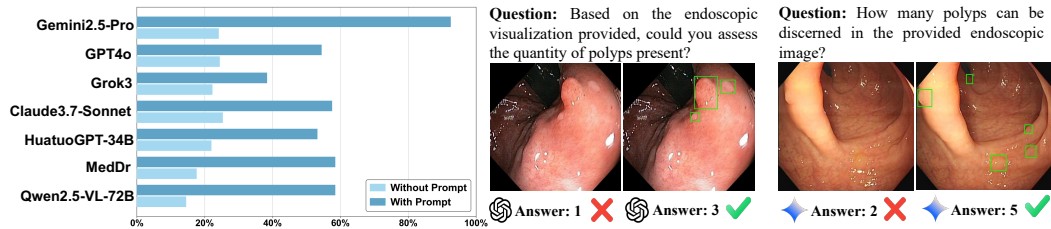

Figure 5: The influence of visual prompt in lesion quantification task among different MLLMs.

even outperforming all proprietary models. This indicates that domain pretraining effectively equips models with essential medical knowledge, enhancing their competitiveness in specialized tasks. However, some medical models exhibit limitations in instruction-following capabilities and suffer from overfitting, which restricts their performance in broader application scenarios. This suggests that while conducting domain-specific training, greater attention should be paid to balancing model generalization and task adaptability.

**3) Model performance varies with visual prompt formats, exposing a gap between visual perception and medical comprehension.** The ability of models to understand spatial information varies significantly based on how visual prompts are formatted. To explore this, we test the same images across 3 tasks with different visual prompts, and the results in Table 2 and Table 3 reveal that most models, especially proprietary ones, excelled in the Region Selection task, indicating strong visual comprehension in distinguishing between regions. However, they struggle to accurately classify lesion types within those regions, pointing to a lack of medical knowledge as the main source of errors rather than poor visual processing. A case study is shown in Fig. 4, and it suggests that while models can spatially differentiate key areas, their interpretation hinges on both the prompt format and their insufficient medical knowledge.

**4) Polyp counting exposes dual challenges in lesion identification and numerical reasoning.** Initial testing reveals severe limitations in this task, with no model achieving above 30% accuracy. To better understand these performance issues, we add bounding boxes as visual prompts (Fig. 5), which dramatically improve accuracy across all models. Most notably, Gemini-2.5-Pro achieves 92.57% from 24.46% with this new prompting approach. This improvement suggests that while Gemini possesses robust spatial recognition and counting abilities, the primary challenge for models lies not in computational or spatial reasoning but in lesion identification. Our findings highlight the importance of incorporating domain-specific medical knowledge into MLLMs to enhance their performance in tasks that combine visual analysis with clinical expertise.

## 5 Conclusion

We introduce EndoBench, the most comprehensive benchmark to date for evaluating multi-modal large language models in endoscopic image analysis. Our results show that while proprietary and domain-adapted MLLMs outperform open-source models in many tasks, all models still fall significantly short of human clinician performance—especially in complex, nuanced clinical scenarios. EndoBench exposes key limitations in current MLLMs' clinical dignaosis and spatial understanding, highlighting the need for further research in domain adaptation and prompt design. We hope EndoBench will serve as a valuable resource for advancing clinically relevant AI in endoscopy.

# 6 Acknowledgments

This work was supported by the Hong Kong Research Grants Council (RGC) General Research Fund under Grant 14204321,14220622, and Hong Kong Innovation and Technology Commission Innovation and Technology Fund PRP/082/24FX.

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

**EndoBench: A Comprehensive Evaluation of Multi-Modal Large Language Models for Endoscopy Analysis**

## Appendix

# A Dataset Details

## A.1 Involved Datasets

In this work, we gather 20 public endoscopy datasets from online sources to encompass various endoscopic image types and professional terminology, including Kvasir [73], HyperKvasir [71], Kvasir-Capsule [74], GastroVision [75], KID [76], WCEBleedGen [77], SEE-AI [78], Kvasir-Seg [79], CVC-ColonDB [80], ETIS-Larib [81], CVC-ClinicDB [82], CVC-300 [83], EDD2020 [84], SUN-Database [85], LDPolypVideo [86], PolypGen [87], Cholec80 [69], EndoVis-17 [88], EndoVis-18 [70], and PSI-AVA [68]. On the other hand, we further enhance the data diversity by incorporating a private wireless capsule endoscopy image dataset from partner hospitals. All data undergoes the privacy de-identification in accordance with medical ethics requirements.

Table 4: Statistics regarding the endoscopic scenarios and dataset information covered by the datasets involved.

| Index | Name | Scenario | Num | Task | Access |
|-------|------|----------|-----|------|--------|
| 1 | Kvasir | GS, CS, SE | 8000 | Classification | Open Access |
| 2 | HyperKvasir | GS, CS, SE | 10662 | Classification | Open Access |
| 3 | Kvasir-Capsule | CE | 47238 | Classification | Open Access |
| 4 | GastroVision | GS, CS, SE | 8000 | Classification | Open Access |
| 5 | KID | CE | 2371 | Classification, Segmentation | Restricted Access |
| 6 | WCEBleedGen | CE | 2618 | Classification, Segmentation | Open Access |
| 7 | SEE-AI | GS | 18481 | Classification | Open Access |
| 8 | Kvasir-Seg | CS | 1000 | Segmentation | Open Access |
| 9 | CVC-ColonDB | CS | 380 | Segmentation | Open Access |
| 10 | ETIS-Larib | CS | 196 | Segmentation | Open Access |
| 11 | CVC-ClinicDB | CS | 612 | Segmentation | Open Access |
| 12 | CVC-300 | CS | 60 | Segmentation | Open Access |
| 13 | EDD2020 | CS | 386 | Classification, Segmentation | Open Access |
| 14 | SUN-Database | CS | 130519 | Classification, Segmentation | Restricted Access |
| 15 | LDPolypVideo | CS | 40266 | Detection | Open Access |
| 16 | PolypGen | CS | 8037 | Segmentation | Open Access |
| 17 | Cholec80 | SE | 8080 | Classification, Segmentation | Open Access |
| 18 | EndoVis-17 | SE | 2235 | Classification, Segmentation | Open Access |
| 19 | EndoVis-18 | SE | 2400 | Classification, Segmentation | Open Access |
| 20 | PSI-AVA | SE | 4471 | Classification, Segmentation | Open Access |
| 21 | WCE2025 | CE | 23447 | Classification | In House |

Abbreviation: GS for Gastroscopy, CS for Colonoscopy, CE for Capsule endoscopy, SE for Surgical endoscopy.

WCE2025 is a capsule endoscopy dataset, meticulously curated with all privacy information removed and approved for public use through agreements with relevant hospitals. The dataset, annotated by professional medical experts, comprises a total of 23,447 samples and is designed for three distinct tasks: Landmark Identification with 1,665 samples, Lesion Type Identification with 19,881 samples, and Organ Identification with 1,901 samples.

## A.2 Task Definition

To comprehensively evaluate these capacities in MLLMs, EndoBench encompasses 12 clinical tasks with 12 secondary subtasks across 4 major categories for endoscopy analysis. These categories include: (1) *anatomical structure recognition* (organ identification, landmark identification); (2) *lesion analysis and grading* (lesion quantification, lesion type identification, lesion severity grading); (3) *spatial localization and region understanding* (visual grounding, region selection, region recognition); and (4) *surgical workflow and operation analysis* (preoperative assessment, macro phases identification, micro operation analysis, instrument management).

### A.2.1 Anatomical Structure Recognition

Anatomical structure recognition is a critical component of endoscopic analysis, enabling the identification of anatomical features within the gastrointestinal tract or other internal structures. This category focuses on two key subtasks:

- **Landmark Identification (LI).** This task involves detecting and labeling specific anatomical landmarks, such as the pylorus, cardia, or ileocecal valve, which serve as reference points during endoscopic navigation. Accurate landmark identification ensures precise orientation and facilitates diagnostic accuracy.

- **Organ Identification (OI).** This task requires recognizing and classifying entire organs or organ segments visible in endoscopic images, such as the esophagus, stomach, or small/large intestine. Organ identification is essential for contextualizing findings and guiding subsequent clinical decisions.

### A.2.2 Lesion Analysis and Grading

Lesion analysis and grading focus on characterizing abnormalities observed during endoscopy, providing critical information for diagnosis and treatment planning. This category encompasses the following subtasks:

- **Lesion Quantification (LQ).:** This task mainly involves measuring the number of polyps, which is crucial for assessing the extent of disease, monitoring progression, and guiding treatment decisions. Accurate quantification provides essential data for evaluating health risks and planning effective interventions.

- **Lesion Type Identification (LT):** This task includes:
  - *Lesion Classification (LC).:* Categorizing lesions into broad types, such as erosions, ulcers, esophagitis, or angiectasia, based on their visual characteristics.
  - *Polyp Type Classification (PT).* Specifically identifying polyp types, such as adenomatous, hyperplastic, or serrated, which have distinct clinical implications.

- **Lesion Severity Grading (LS).** This task involves assessing the severity of ulcerative colitis through colonoscopy, using the UCEIS Mayo Score to classify mucosal inflammation into four levels (0: inactive, 1: mild, 2: moderate, 3: severe) and additional in-between classes (0-1, 1-2, 2-3) to account for observer variation and nuanced disease presentations. This task is critical for determining disease extent, guiding treatment, and monitoring progression.

### A.2.3 Surgical Workflow and Operation Analysis

Surgical workflow and operation analysis focus on understanding and optimizing endoscopic procedures, from preoperative planning to intraoperative management. This category includes the following tasks:

- **Preoperative Assessment (PA).** This task focuses on evaluating bowel cleanliness prior to surgical or endoscopic procedures, primarily through the Boston Bowel Preparation Scale (BBPS). This task involves scoring the quality of mucosal views in endoscopic images or videos, using only the BBPS 0-1 (poorly prepared, with significant stool or fluid obstructing the view) and BBPS 2-3 (well-prepared, with minimal or no residue, including perfectly clean BBPS 3 cases) classes, with the impacted stool class excluded.

- **Macro Phases Identification (MP):** This task includes:
  - *Surgical Phase Recognition (SP).* Identifying distinct phases of an endoscopic procedure, such as clipping and cutting, gallbladder dissection, preparation, gallbladder retraction, and others. This task is essential for enhancing procedural efficiency, supporting surgical training, and enabling automated analysis of surgical workflows.
  - *Surgical Step Recognition (SS).* Identifying specific steps within an endoscopic procedure phase, such as inserting the prostate into a retrieval bag, prostate dissection until the levator ani, or vascular pedicle control. This task is essential for tracking procedural progress, enhancing surgical precision, and supporting training and automated surgical workflow analysis.

- **Micro Operation Analysis (MO).** This task includes:

- *Surgical Action Recognition (SA).* Detecting specific actions performed by a particular instrument during endoscopic interventions, such as suction, tissue manipulation, suturing, or idle states. This task is crucial for analyzing instrument-specific activities, optimizing procedural efficiency, and supporting surgical training and automation.

- *Treatment Intervention Recognition (TI).* This task involves identifying and classifying therapeutic interventions performed during endoscopic procedures to address detected anomalies, such as lesion or polyp removal. It includes recognizing actions like polyp resection, biopsy of resection margins or sites, and the use of indigo carmine injection to enhance lesion demarcation, where the blue color beneath the dyed, lifted polyp highlights accurate polyp margins. This task is critical for ensuring precise treatment execution and supporting procedural documentation and analysis.

- **Instrument Management (IM).** This task includes:

  - *Instrument Counting (IC).* This subtask involves identifying and counting the number of distinct types of instruments used during an endoscopic procedure. It requires recognizing various instruments present in the endoscopic images or videos to provide an accurate count of different tools, such as forceps, scissors, or suction devices.

  - *Instrument Presence Verification (IP).* This subtask involves determining whether a specific instrument, such as forceps, scissors, or a suction device, is present in endoscopic images or videos. It requires analyzing the visual data to confirm the presence or absence of the specified instrument, which is crucial for ensuring appropriate tool usage, tracking procedural steps, and supporting surgical workflow management.

### A.2.4 Spatial Localization and Region Understanding

Spatial localization and region understanding enable precise mapping and interpretation of regions of interest within endoscopic images. This category includes the following subtasks:

- **Visual Grounding (VG).** This task involves associating textual descriptions or clinical queries with specific regions in endoscopic images to accurately identify relevant features, such as anatomical structures, lesions, or abnormalities. The task requires selecting the correct coordinate location from four candidate options to ensure precise localization of these features in the images.

- **Region Selection (RS)**: This task involves identifying and selecting key regions of interest, such as abnormal tissue areas or anatomical structures, in endoscopic images for further analysis or intervention. The task requires choosing the correct region from four candidate regions, each marked with a different color, to ensure accurate localization of the critical area.

- **Region Recognition (RR).** This task involves identifying and classifying specific structures or lesions in endoscopic images using various visual prompt methods to determine what a given region represents. The task includes the following subtasks:

  - *Bounding Box Region Recognition.* A rectangular bounding box is overlaid on the endoscopic image to highlight a specific area, and the task is to identify the structure or lesion within that boxed region.

  - *Contour-based Region Recognition.* Precise boundary outlines are drawn on the image to determine the corresponding structure or lesion.

  - *Multi-region Recognition.* Multiple regions within a single endoscopic frame are marked with distinct colors, and the task is to identify the structure or lesion associated with a specified color.

  - *Coordinate-based Region Recognition.* Specific coordinates (e.g., $[x1, y1, x2, y2]$) are provided in the question, and the task is to identify the structure or lesion at that precise location, facilitating integration with coordinate-based systems.

### A.2.5 Summarization of Clinically-Grounded Tasks

The 12 tasks in EndoBench are systematically designed to mirror the end-to-end clinical workflow of an endoscopic procedure, which encompasses three critical phases: diagnostic assessment, surgical planning, and therapeutic implementation. Each task is mapped to a specific clinical need within this progression.

The workflow begins with the diagnostic assessment phase. When a patient undergoes an endoscopic evaluation, the procedure involves a systematic examination of the gastrointestinal tract. Organ Identification provides real-time anatomical orientation as the endoscope navigates between the esophagus, stomach, and duodenum. Concurrently, Landmark Identification recognizes critical structures, such as the pylorus or ampulla of Vater, to ensure a complete and thorough examination. Upon detecting an abnormality, Lesion Type Identification characterizes its pathology (e.g., polyps, ulcers, inflammation), Lesion Quantification counts multiple instances for treatment planning, and Lesion Severity Grading applies standardized scoring systems (e.g., Mayo Score, BBPS) to guide therapeutic decisions.

Following diagnosis, the process transitions to the surgical planning phase, where detailed preparation for intervention is critical. In this stage, Visual Grounding provides precise coordinates for surgical navigation systems, Region Selection facilitates accurate target localization, Region Recognition characterizes specific anatomical structures within an area of interest, and Preoperative Assessment evaluates key safety parameters before an intervention begins.

The final stage is therapeutic implementation, where the focus shifts to real-time procedural execution and quality assurance. Macro Phases Identification offers high-level workflow guidance by recognizing distinct surgical stages (e.g., dissection, resection, suturing). To enhance patient safety, Instrument Management monitors the presence of tools to prevent retained equipment. Finally, Micro Operation Analysis provides a granular assessment of surgical skill and quality by analyzing specific instrument functions, such as grasping, cutting, and cauterization.

By structuring the tasks around this comprehensive workflow, EndoBench ensures that MLLMs are evaluated on capabilities that directly translate to improved patient care, enhanced surgical navigation, and more informed clinical decision-making in real-world endoscopic practice. This end-to-end mapping underscores the clinical value and practical relevance of the benchmark.

### A.3 Construction Process of QA Pairs

**Task Templates.** To ensure a comprehensive and varied evaluation of model performance, we develop 5 to 8 distinct question templates for each task. These templates are designed to cover a range of scenarios and complexities, enabling robust testing of the model's capabilities. The table below outlines the specific prompts associated with each task, providing a structured framework for generating diverse and targeted questions.

---

**Prompt for Answer Evaluation**

```
Landmark Identification:
What anatomical landmark is highlighted in this image?
Which anatomical landmark is visible in this image?
Can you identify the anatomical landmark in this image?
Identify the anatomical landmark in this image.
What is the name of the anatomical landmark in this image?
Which anatomical structure is shown in this image?
Can you identify the anatomical structure in this image?
Identify the anatomical structure in this image.
What is the name of the anatomical feature marked in this image?

Organ Identification:
What organ is shown in this image?
Which part of the digestive system is depicted in this image?
Can you identify the organ in this image?
This image shows characteristic features of which digestive organ?
Can you identify the digestive organ in this image?
Identify the digestive organ in this image.
What is the name of the digestive organ shown in this image?

Lesion Quantification:
Based on the image provided, how many polyps are present in the image?
Can you identify the number of polyps in this image?
```

---

Please identify the number of polyps shown in the image.
Given the endoscopic image, can you determine the number of polyps?
Identify the number of polyps in this endoscopic image.

Lesion Classification:
Is there any abnormality visible in this image?  If so, describe the type of
abnormality.
Based on this endoscopic image, what type of abnormal finding can be
identified?
Does this endoscopic image show any abnormalities?  If yes, please specify the
type.
Are there any abnormal findings in this image?  If present, what type of
abnormality is it?
Please examine this image and indicate if there are any abnormalities.  If so,
what kind?
Review this image and state if there are any abnormalities.  If found, specify
the type.
Check this image for any abnormalities.  If detected, what type of abnormality
is present?
Analyze this image and report if there are any abnormalities.  If yes,
describe the type.
Evaluate this image for abnormalities.  If any are found, what type are they?

Lesion Severity Grading:
What is the Mayo Score for ulcerative colitis in this endoscopic image?
Can you determine the Mayo Score for ulcerative colitis in this endoscopic
image?
Based on the Mayo Score, what score would you give this endoscopic image for
ulcerative colitis?
According to the Mayo Score, how would you score this endoscopic image for
ulcerative colitis?
What score does this endoscopic image achieve when assessed using the Mayo
Score for ulcerative colitis?

Polyp Type Classification:
Based on the image provided, identify the histopathological type of the
lesion.
Can you identify the histopathological type of the lesion in this image?
Please identify the histopathological type of the lesion shown in the image.
Given the endoscopic image, can you determine the colorectal lesion type?
What type of colorectal lesion is depicted in the image?

Visual Grounding:
Could you give the location of the {lesion_type} in this endoscopic image?
Please specify the coordinates of the {lesion_type} in this endoscopic image.
Could you specify the location of the {lesion_type} in this endoscopic image?
Please give the coordinates of the {lesion_type} in this endoscopic image.
Please specify the location of the {lesion_type} in the image.
Could you identify the coordinates of the {lesion_type} in the image?

Region Selection:
In the given image, which color box best represents the area of the
{lesion_type}?
In the provided image, which color box best indicates the location of the
{lesion_type}?
In this endoscopic image, which color box best highlights the {lesion_type}?
Which color box in the image best describes the {lesion_type}?
Which color box in this endoscopic image best represents the {lesion_type}?

Bounding Box Region Recognition:
Which option best describes the region marked by the rectangle in the
endoscopy image?

In the endoscopy image, which option best describes the region marked by the rectangle?
Which option best describes the region highlighted by the rectangle in the endoscopy image?
In this endoscopy image, which option best describes the highlighted region marked by the rectangle?
In this endoscopy image, which surgical instrument is indicated by the {color} bounding box?
Given the endoscopy image, what surgical instrument is shown in the {color} bounding box?
Based on the endoscopy image, identify the surgical instrument in the {color} bounding box.
Which surgical instrument corresponds to the {color} bounding box in this endoscopy image?
Determine the surgical instrument in the {color} bounding box from the endoscopy image.

Contour-based Region Recognition:
Which option best describes the region marked by the contour in the endoscopy image?
In the endoscopy image, which option best describes the region marked by the contour?
Which option best describes the region highlighted by the contour in the endoscopy image?
In this endoscopy image, which option best describes the highlighted region marked by the contour?

Multi-region Recognition:
Which option best describes the region marked by the {color} rectangle in the endoscopy image?
In the endoscopy image, which option best describes the region marked by the {color} bounding box?
Which option best describes the region highlighted by the {color} rectangle in the endoscopy image?
In this endoscopy image, which option best describes the highlighted region marked by the {color} bounding box?
Which option best describes the region marked by the {color} contour in the endoscopy image?
In the endoscopy image, which option best describes the region marked by the {color} contour?
Which option best describes the region highlighted by the {color} contour in the endoscopy image?
In this endoscopy image, which option best describes the highlighted region marked by the {color} contour?

Coordinate-based Region Recognition:
Could you identify the findings in location at [{x1}, {y1}, {x2}, {y2}] in this endoscopic image?
What type of finding can be identified at [{x1}, {y1}, {x2}, {y2}] in this endoscopic image?
Please specify the findings at [{x1}, {y1}, {x2}, {y2}] in this endoscopic image.
Could you specify the findings at [{x1}, {y1}, {x2}, {y2}] in this endoscopic image?
Analyze this image and specify the findings at [{x1}, {y1}, {x2}, {y2}].
What surgical instrument is located within the bounding box at coordinates [{x1}, {y1}, {x2}, {y2}] in this endoscopy image?
Identify the surgical instrument inside the rectangle at coordinates [{x1}, {y1}, {x2}, {y2}] in this endoscopy image.
Which surgical instrument is within the coordinates [{x1}, {y1}, {x2}, {y2}] in this endoscopy image?
In this endoscopy image, what is the surgical instrument at the bounding box

[{x1}, {y1}, {x2}, {y2}]?
Determine the surgical instrument located at coordinates [{x1}, {y1}, {x2}, {y2}] in this endoscopy image.

Preoperative Assessment:
What is the Boston Bowel Preparation Scale (BBPS) score for this endoscopic image?
Can you determine the score for this endoscopic image based on the Boston Bowel Preparation Scale?
Based on the Boston Bowel Preparation Scale, what score would you give this endoscopic image?
According to the Boston Bowel Preparation Scale, how would you score this endoscopic image?
What score does this endoscopic image achieve when assessed using the Boston Bowel Preparation Scale?

Surgical Phase Recognition:
This is an endoscopy image.  Which surgical phase is currently being performed?
Given this endoscopy image, can you identify the ongoing surgical phase?
Based on the endoscopy image provided, what surgical phase is depicted?
Looking at this endoscopy image, which surgical phase does it correspond to?
In the context of this endoscopy image, determine the current surgical phase.

Surgical Step Recognition:
Given this endoscopy image.  Which surgical step is being performed?
In the endoscopy image, what surgical step is currently underway?
Based on this endoscopy image, can you identify the surgical step?
Which surgical step does this endoscopy image correspond to?
From the endoscopy image provided, determine the surgical step.

Surgical Action Recognition:
In this endoscopy image, what is the state of the {instrument}?
What surgical action is the {instrument} performing in this endoscopy image?
Identify the state of the {instrument} in this endoscopy image.
What is the {instrument} doing in this endoscopy image?
Determine the action of the {instrument} in this endoscopy image.

Treatment Intervention Recognition:
What is the therapeutic intervention in this endoscopic image?
Can you identify the therapeutic intervention in this image?
Identify the therapeutic intervention in this image.
Which therapeutic intervention is shown in this image?
Which therapeutic intervention is performed in this image?

Instrument Counting:
This is an endoscopy image.  How many distinct types of surgical instruments can be identified?
In the endoscopy image provided, which option correctly states the number of unique surgical instrument categories?
Based on this endoscopy image, how many different classifications of surgical instruments are visible?
From the endoscopy image, can you determine the number of unique surgical instrument varieties present?
Which option accurately reflects the count of distinct surgical instrument types in this endoscopy image?

Instrument Presence Verification:
In this endoscopy image, is {instrument} present during {phase}?
Based on this endoscopy image, is {instrument} used in {phase}?
Does this endoscopy image show {instrument} during {phase}?
In the context of this endoscopy image, is {instrument} visible in {phase}?

```
Is {instrument} present in this endoscopy image during {phase}?
```

**Refined QA.** To improve dataset variety and thoroughly assess the capabilities of Multimodal Large Language Models (MLLMs), we employ the GPT-4o-mini API to rephrase questions from the original QA pairs. The following is the prompt provided to GPT-4o-mini:

**Prompt for Refined QA**

```
Please rewrite the following question text to make its expression more
diverse, while keeping the core meaning unchanged.  The question pertains
to an endoscopic image, so please incorporate knowledge from the medical
field.  Use varied sentence structures and appropriate synonyms, avoiding
direct repetition of the original sentence.
If possible, include professional expressions commonly used in the medical
domain.
If you need a reference example, here is a sample question and its rewritten
version:
Original question:
'What is the lesion in this endoscopic image?'
Rewritten question:
'What type of abnormality might the region observed in this endoscopic image
represent?'
Now, please rewrite the following question text:
The original question text is as follows:
{report}
and the answer options are as follows:
{options}
Please only give one rewritten version of the question like <question> the
rewritten question</question>, and DO NOT add any other content like options
or answers.
```

## A.4 Well-categorized Data Structure

In this work, we construct EndoVQA-Instruct dataset, yielding 446,535 VQA pairs, which is the current largest endoscopic instruction-tuning collection. Fig. 6 shows the distribution of the EndoVQA-Instruct dataset. The dataset exhibits an imbalanced class distribution, with normal samples dominating the Lesion Type Identification task. To address this issue and ensure a more robust evaluation, a smaller, curated subset of the dataset is selected. From EndoVQA-Instruct, we extract representative pairs that undergo rigorous clinical review, resulting in our final EndoBench dataset of 6,832 clinically validated VQA pairs. The construction pipeline is shown in Section 3.1.

Regarding surgical procedures, the surgical endoscopy subset includes four representative surgery types:

- Abdominal porcine surgeries (30.77%) from EndoVis17 and EndoVis18,
- Human laparoscopic cholecystectomy (16.05%) from Cholec80,
- Human radical prostatectomy (41.98%) from PSI-AVA, and
- Endoscopic mucosal resection (11.20%) from Kvasir, HyperKvasir, and GastroVision.

## A.5 EndoBench-Extended

To evaluate model performance on rare and complex clinical scenarios—critical for assessing robustness and real-world applicability—we introduce EndoBench-Extended, an extended version of the original EndoBench dataset. EndoBench-Extended comprises 48 endoscopic images and 91 open-set VQA pairs, all carefully curated by medical experts to ensure high clinical relevance and diagnostic challenge. The dataset specifically targets underrepresented and difficult cases, including rare pathologies, overlapping or ambiguous lesions, atypical anatomical structures, and post-surgical endoscopic appearances. By focusing on these clinically nuanced endoscopic scenarios, EndoBench-Extended

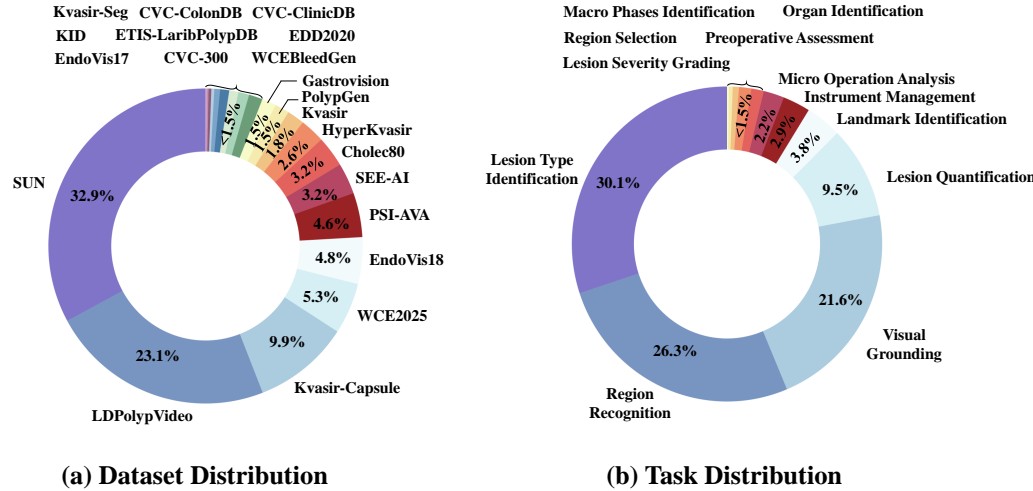

(a) Dataset Distribution      (b) Task Distribution

Figure 6: Data distribution of the EndoVQA-Instruct dataset.

serves as a valuable benchmark for both quantitative evaluation and subjective assessment of AI models in gastrointestinal endoscopy and related clinical settings.

# B   Evaluation

## B.1   Evaluated Models

This study evaluates 23 distinct Multi-modal Large Language Models (MLLMs), spanning open-source architectures, domain-specific medical models, and closed-source general-purpose models accessed via proprietary APIs.

- **LLaVA [32]**: This series employs an end-to-end training framework, integrating CLIP-based [94] vision encoders with large language models for robust visual and linguistic understanding. LLaVA-v1.5-7B [32] and LLaVA-v1.5-13B [32] use a Vicuna [95] backbone, achieving strong performance on benchmarks like Science QA [96]. LLaVA-llama3-8B [32] and LLaVA-Next-Llama3-8B [90], built on the 8-billion-parameter LLaMA-3 model [97], enhance visual reasoning and OCR capabilities for diverse multimodal applications.

- **CogVLM-Chat-7B [33]**: Combining a vision transformer encoder, an MLP adapter, and a visual expert module, this model enables deep fusion of visual and linguistic features, excelling in tasks like image captioning and visual question answering with a pretrained language model backbone.

- **ShareGPT-4v-7B [34]**: This open-source chatbot is trained by fine-tuning a CLIP vision tower and LLaMA/Vicuna on GPT4-Vision-assisted ShareGPT4V data [98] and LLaVA instruction-tuning data. Leveraging the ShareGPT4V dataset, it captures detailed visual information, including world knowledge and spatial relationships, enhancing performance across multimodal benchmarks through supervised fine-tuning.

- **Qwen2.5VL [41]**: With 3B, 7B, and 72B parameter variants, this series introduces dynamic resolution and frame-rate sampling for video understanding. It excels in object localization, structured output generation for documents like invoices, and advanced visual recognition, supported by an optimized vision encoder with window attention.

- **Janus-Pro-7B [36]**: This model employs a novel autoregressive framework that unifies multimodal understanding and generation. It addresses the limitations of previous approaches by decoupling visual encoding into separate pathways while utilizing a single, unified transformer architecture. Using a SigLIP-L encoder [99] for 384x384 image inputs and a specialized tokenizer for generation, it supports efficient high-resolution image processing.

- **InternVL2.5 [38]**: Comprising 8B and 38B parameter variants, this series enhances multimodal capabilities with optimized training strategies and high-quality data, supporting complex visual-language interactions and high-resolution image processing.

- **QvQ-72B [91]**: Focused on visual reasoning, this experimental model achieves a 70.3% score on the MMMU benchmark [100], excelling in multidisciplinary understanding, mathematical reasoning, and Olympiad tasks, supporting single-round dialogues and image outputs.

- **MedDr-80B [48]**: This model handles diverse medical imaging modalities, including radiology, pathology, and endoscopy, using a diagnosis-guided bootstrapping strategy to create high-quality datasets, boosting performance in visual question answering and medical report generation.

- **LLaVA-Med-7B [44]**: A biomedical adaptation of LLaVA, this model is fine-tuned with a curriculum learning approach, aligning biomedical vocabulary with figure-caption pairs for efficient handling of medical image queries and multimodal conversations.

- **HuatuoGPT-Vision [47]**: Available in 7B and 34B variants, this series leverages the PubMed-Vision dataset [47] with 1.3 million medical VQA samples, significantly improving medical multimodal benchmarks and enhancing clinical visual understanding.

- **ColonGPT [13]**: Specialized for colonoscopic image analysis, this model integrates a SigLIP vision encoder and a compact language model, enabling precise diagnostics in gastroenterology through multimodal processing.

- **DeepSeek-V3 [101]**: Built on a Mixture-of-Experts architecture with 671B parameters, this proprietary model delivers efficient inference for high-resolution image processing and complex reasoning across multimodal tasks.

- **Grok-3 [93]**: Designed for versatile multimodal interactions, this model supports text and image inputs with robust reasoning capabilities, optimized for real-time performance across multiple platforms.

- **Claude-3.7-Sonnet [43]**: A hybrid reasoning model, it excels in content generation, data analysis, and visual reasoning, offering advanced capabilities for diverse multimodal and clinical applications.

- **GPT-4o [92]**: A flagship multimodal model, it seamlessly integrates text and image processing, supporting a wide range of tasks requiring robust visual and linguistic understanding.

- **Gemini-2.5-Pro [102]**: Based on a sparse Mixture-of-Experts Transformer, this model enhances complex reasoning, processing inputs from text, images, and code repositories with improved training stability and performance.

## B.2 Detailed Setup

All model weights are obtained from their official repositories on Hugging Face to ensure consistency and reliability. The evaluation is carried out under a zero-shot learning paradigm, where no task-specific training data or in-context examples are provided to the models.

We adopt a closed-set evaluation protocol to facilitate systematic performance assessment. In this setup, each task is presented as a multiple-choice question, where models are required to select the correct answer from a predefined set of options. The adoption of a closed-set evaluation is particularly well-suited to the medical domain, where diagnostic reasoning often involves distinguishing between highly similar conditions or subtly differentiated concepts. Open-ended generation in such contexts may lead to semantically plausible yet technically incorrect responses, making it difficult to assess precise comprehension. Predictions are evaluated using exact matching against ground truth labels, and we report the accuracy at the task-wise and overall levels. The following is the prompt for evaluation:

**Prompt for Evaluation**

```
USER: <image>
{Question}:
A. {optionA}
B. {optionB}
...
```

```
  Please select the correct answer from the options above.
  ASSISTANT:
```

During the evaluation, we observe that certain MLLMs, especially those tailored for medical applications, struggle with the following instructions, often failing to generate responses in the expected format (e.g., selecting an option letter). This does not necessarily reflect a lack of domain knowledge, but rather a limitation in processing structured question formats. To address this, we employ Qwen2.5VL-72B [41] as an auxiliary judger to extract the most plausible answer from the model's response when the intended choice is ambiguous or missing. If no valid answer can be identified, the sample is treated as an error. The following is the prompt for evaluation:

**Prompt for LLM Judger**

```
You are tasked with evaluating the correctness of a model's output by
comparing it to the ground-truth answer.  Extract a plausible answer from
the model's output.  If no valid answer can be extracted, mark the output as
incorrect.  Compare the extracted answer (or the original output if extraction
is unnecessary) with the ground-truth answer for an exact match.
Output:  Respond with "Yes" if the model's answer exactly matches the
ground-truth answer, or "No" if it does not match or no valid answer could
be extracted.
Model Output:  {model_output}
Ground-Truth Answer:  {ground_truth_answer}
Provide your response as either "Yes" or "No".
```

**Human Study.** To establish a benchmark for performance, the study includes an evaluation of human clinicians. We randomly select 255 questions from our EndoBench across all the sub-tasks except coordinate-related tasks, due to the precise coordinate format being unsuitable for intuitive human judgment. Each sub-task may include 5, 10, or 15 samples. Two certified clinicians with expertise in endoscopy independently finished the selected questions. Their scores are averaged for each task to provide a reference standard for comparison.

## B.3 Additional Results

In this section, we will provide the complete quantitative results of our experiments.

Table 5 presents the performance of various MLLMs across 12 subtasks within the EndoBenchframework. Notably, in the Polyp Type Classification task, ColonGPT [13] significantly outperforms other models, achieving an impressive accuracy of 57.60%. In contrast, competing models exhibit suboptimal performance, primarily due to their limited domain-specific knowledge in this area.

Fig. 7 and 8 illustrate the performance comparisons of existing MLLMs across 4 endoscopic scenarios and 5 levels of visual prompting granularity within the EndoBench, respectively. The results reveal that open-source general-purpose models tend to perform relatively better on surgical images compared to medical-domain-specific models and proprietary models, where the performance advantage is less pronounced. However, across all endoscopic scenarios and visual prompting granularities, a consistent performance gap exists between all evaluated models and expert physicians. This observation underscores the challenges in achieving expert-level proficiency in medical image analysis and highlights the need for further advancements in model design and training to bridge this gap.

## B.4 Case Study

In this section, we conduct a case study analysis of multiple MLLMs in EndoBench under various scenarios.

**Correct Samples.** These figures (Fig. 9-14) highlight exemplary performances by leading models such as Gemini-2.5-Pro [102] and GPT-4o [42]. These models demonstrate robust capabilities in

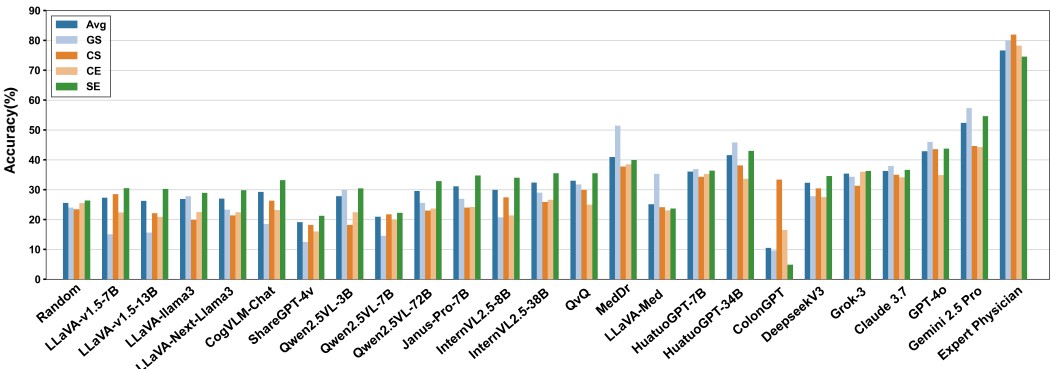

Figure 7: Performance comparison across 4 endoscopic scenarios in EndoBench among existing MLLMs.

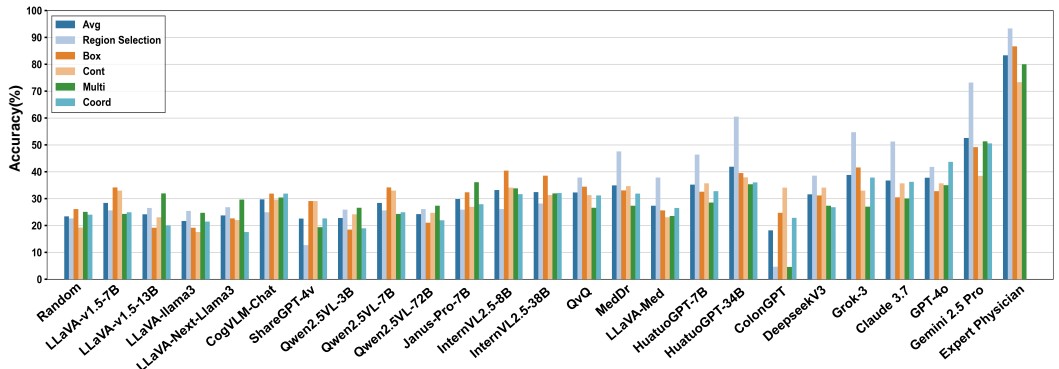

Figure 8: Performance comparison across 5 different visual prompts in EndoBench among existing MLLMs.

accurately interpreting endoscopic images and providing clinically relevant responses, highlighting their potential for assisting in real-world endoscopic analysis.

**Error Analysis.** The errors observed in the case studies are categorized into four types, each highlighting distinct limitations in the performance of multimodal large language models in medical applications.

- **Perceptual Errors:** MLLMs may struggle to accurately perceive or interpret visual information in images, including failing to detect critical objects, misidentifying elements, or overlooking essential details. In Fig. 15, QvQ-72B [91] fails to recognize erythematous areas and focuses on irrelevant yellow-white granules. Similarly, in Fig. 16, HuatuoGPT-Vision-34B [47] overlooks that the mucosa has been stained blue, leading to an incorrect interpretation of the scene. These indicate a limitation in the model's ability to accurately recognize clinically significant visual patterns.

- **Lack of Knowledge:** MLLMs may accurately identify visual elements in an image and comprehend the question but still provide incorrect answers due to insufficient medical domain expertise. This manifests as misinterpretations of clinical signs or failure to differentiate between similar medical conditions. For instance, in Fig. 17, QvQ-72B [91] correctly identifies low-level visual features, such as red points in the image, but misinterprets them as blood vessels. Similarly, in Fig. 18, HuatuoGPT-Vision-34B [47] notices prominent bright red areas in the image during reasoning but fails to interpret them as bleeding, leading to an inaccurate response. These errors highlight a deficiency in domain-specific medical knowledge, where the model fails to contextualize visual cues with appropriate clinical understanding.

- **Irrelevant Response:** MLLMs sometimes generate responses that are unrelated to the user's query, producing irrelevant, incomplete, or incomprehensible information that fails to address the

Table 5: Results of different MLLMs on 12 subtasks in EndoBench. The best-performing model in each category is **in-bold**, and the second best is underlined.

| MLLMs | LC | PT | SP | SS | SA | TI | IC | IP | Box | Cont | Mul | Coor |
|---|---|---|---|---|---|---|---|---|---|---|---|---|
| Random | 24.50 | 19.20 | 25.71 | 25.52 | 26.60 | 29.18 | 25.63 | 48.98 | 26.10 | 19.23 | 25.10 | 24.02 |
| Physician | 86.67 | 70.00 | 53.33 | 66.67 | 85.00 | 66.67 | 90.00 | 73.33 | 86.67 | 73.33 | 80.00 | - |
| **Open-Source MLLMs** | | | | | | | | | | | | |
| Llava-v1.5-7B | 21.29 | 22.80 | 24.86 | 23.71 | 54.29 | 23.61 | 30.15 | 46.94 | 34.18 | 32.97 | 24.33 | 24.94 |
| Llava-v1.5-13B | 19.68 | 20.00 | 19.43 | 25.00 | 55.12 | 36.39 | 29.15 | 45.92 | 19.17 | 23.08 | 31.94 | 20.09 |
| Llava-llama3-8B | 25.25 | 13.20 | 23.43 | 23.97 | 50.14 | 33.44 | 31.66 | 47.96 | 19.17 | 17.58 | 24.71 | 21.48 |
| Llava-Next-Llama3-8B | 17.70 | 14.80 | 24.57 | 24.48 | 53.74 | 34.43 | 37.69 | 43.88 | 22.63 | 21.98 | 29.66 | 17.55 |
| CogVLM-Chat-7B | 21.91 | 18.40 | 21.43 | 25.77 | 57.06 | 22.95 | 27.14 | 50.00 | 31.87 | 29.67 | 30.42 | 31.87 |
| ShareGPT-4v-7B | 17.70 | 24.00 | 13.71 | 18.56 | 45.98 | 14.10 | 29.15 | 6.12 | 29.10 | 29.12 | 19.39 | 22.63 |
| Qwen2.5VL-3B-Instruct | 22.03 | 16.40 | 24.86 | 27.06 | 57.89 | 23.93 | 55.78 | 38.78 | 18.48 | 24.18 | 26.62 | 18.94 |
| Qwen2.5VL-7B-Instruct | 21.29 | 22.80 | 24.86 | 23.71 | 54.29 | 23.61 | **63.32** | 46.94 | 34.18 | 32.97 | 24.33 | 24.94 |
| Qwen2.5VL-72B-Instruct | 22.40 | 21.60 | 32.29 | 23.20 | 56.79 | 37.70 | 54.27 | 41.84 | 21.02 | 24.73 | 27.38 | 21.94 |
| Janus-Pro-7B | 22.90 | 20.40 | 26.29 | 28.35 | 57.89 | 36.07 | 43.22 | 50.00 | 32.33 | 26.92 | 36.12 | 27.94 |
| InternVL2.5-8B | 19.06 | 19.20 | 28.00 | 25.77 | 60.39 | 36.07 | 46.73 | 43.88 | 40.42 | 34.07 | 33.84 | 31.64 |
| InternVL2.5-38B | 23.51 | 17.20 | 30.86 | 24.74 | 59.83 | 33.11 | 51.76 | 45.92 | 38.57 | 31.32 | 31.94 | 32.10 |
| QvQ-72B | 33.42 | 22.80 | 27.14 | 30.41 | 54.57 | 44.26 | 54.77 | 52.04 | 34.41 | 31.32 | 26.62 | 31.18 |
| **Open-Source Medical-Domain MLLMs** | | | | | | | | | | | | |
| MedDr-80B | 42.95 | 17.60 | 26.57 | 29.38 | 58.17 | 56.72 | 44.72 | 56.12 | 33.03 | 34.62 | 27.38 | 31.87 |
| Llava-Med-7B | 12.75 | 25.60 | 17.43 | 17.78 | 16.90 | 34.43 | 29.15 | 17.35 | 25.64 | 23.08 | 23.57 | 26.56 |
| HuatuoGPT-Vision-7B | 41.46 | 24.40 | 29.43 | 32.47 | 52.36 | 46.89 | 44.72 | 59.18 | 32.56 | 35.71 | 28.52 | 32.79 |
| HuatuoGPT-Vision-34B | 39.48 | 20.40 | 32.57 | 38.66 | 57.34 | 52.79 | 59.30 | 59.18 | 39.49 | 37.91 | 35.36 | 36.03 |
| ColonGPT | 18.44 | **57.60** | 11.14 | 0.26 | 0.83 | 2.62 | 16.08 | 15.31 | 24.71 | 34.07 | 4.56 | 22.86 |
| **Proprietary MLLMs** | | | | | | | | | | | | |
| Deepseek-V3 | 33.79 | 17.60 | 24.00 | 31.44 | 60.38 | 42.30 | 55.28 | 53.06 | 31.18 | 34.07 | 27.38 | 26.79 |
| Grok-3 | 39.11 | 16.00 | 28.29 | 26.55 | 58.73 | 38.03 | 50.75 | 60.20 | 41.57 | 32.97 | 27.00 | 37.88 |
| Claude-3.7-Sonnet | 43.07 | 13.20 | 31.43 | 24.48 | 55.68 | 51.48 | 50.25 | 44.90 | 30.48 | 35.71 | 30.04 | 36.26 |
| GPT-4o | **49.38** | 19.20 | 36.00 | 45.88 | 64.54 | 52.79 | **62.81** | **65.31** | 32.79 | 35.71 | 34.98 | 43.65 |
| Gemini-2.5-Pro | 49.01 | 19.20 | **44.29** | **47.94** | **65.10** | **71.15** | 62.81 | 63.27 | **49.19** | **38.46** | **51.33** | **50.58** |

question. For example, in Fig. 19, LLaVA-Med [44] is asked to determine the number of surgical instruments in an endoscopic image but outputs a tautological restatement of the query, lacking any clinical insight. In another case, Fig. 20, ColonGPT [13] is tasked with classifying pathological findings in an endoscopic image but outputs a term unrelated to the provided options and observed pathology. These case studies emphasize the need for improved medical knowledge integration and enhanced perceptual capabilities to bridge the gap between current MLLM performance and clinical requirements.

- **Refusal to Answer:** Certain MLLMs, particularly proprietary ones, are designed to decline responses to questions involving sensitive information, ethical dilemmas, or requiring professional medical advice to ensure safety and compliance. For example, in Fig. 21, GPT-4o [42] is asked to identify the coordinates of a low-grade adenoma in an endoscopic image but states it is unable to provide the coordinates. Likewise, in Fig. 22, Grok-3 [93] is tasked with counting surgical instruments in an endoscopic image but explicitly refuses, citing its inability to process such requests. These cases highlight the need for enhanced technical capabilities and clearer ethical guidelines to balance safety with clinical utility in MLLM responses.

## C  Limitations

While our current work provides a comprehensive benchmark for 2D endoscopic image analysis, this approach has inherent limitations in clinical applicability. Static 2D images cannot capture critical spatial-depth relationships (e.g., polyp morphology assessment) or temporal dynamics (e.g., bleeding source localization), which are essential for accurate diagnosis and surgical planning. These constraints highlight the need to evolve toward 3D endoscopic video analysis, where volumetric reconstruction and motion context could enable transformative applications like real-time surgical navigation, instrument tracking, and dynamic lesion characterization. Future research must address

the computational and annotation challenges of 3D video to achieve clinically viable systems that complement physician decision-making in complex endoscopic procedures.

Moreover, our work still involves a key remaining challenge of integrating AI into complete clinical workflows, particularly when handling ambiguous cases requiring expert physician interpretation. These complex scenarios demand additional rigorous clinical validation and nuanced clinical judgment that current systems cannot fully replicate. Future work should address these limitations through expanded validation studies and improved algorithmic handling of diagnostic uncertainties. The path to reliable clinical implementation requires both technological advances in AI interpretation and careful workflow integration to complement - rather than replace - physician expertise in challenging cases.

Furthermore, to enable reliable AI-assisted endoscopic diagnosis, future research should pursue: (1) multicenter clinical trials to validate performance across diverse populations and settings; (2) development of standardized benchmarks assessing diagnostic accuracy, workflow integration, and clinical utility; and (3) establishment of ethical frameworks addressing data privacy, algorithmic bias, and physician-AI collaboration. These efforts must focus particularly on challenging areas like indeterminate cases requiring human expertise. Only through such rigorous validation and standardization can we ensure these technologies meet clinical needs while maintaining patient safety and upholding ethical standards in medical practice.

# D  Potential Improvement Methodology

Based on the evaluations of EndoBench, existing MLLMs still have a long way to go before they can be applied clinically. To bridge this performance gap, several key methodologies can be explored, ranging from training-free prompting strategies to more intensive model adaptation techniques.

First, advanced prompt engineering offers a direct path to enhancing performance without additional model training. As shown in Table 6, our extended evaluation of Zero-Shot Chain-of-Thought (CoT) prompting [103] reveals its model-dependent efficacy. While it improved accuracy for robust models like GPT-4o and Gemini-2.5-Pro, it degraded the performance of HuatuoGPT-Vision. This suggests that compelling a model to articulate a reasoning path can paradoxically increase the risk of hallucination when its internal knowledge is not sufficiently grounded [104, 105, 106]. More robust prompting techniques, such as Self-Consistency (SC) [107] and Self-Refine [108], which leverage multiple reasoning paths and iterative feedback, or domain-adapted strategies like MedPrompts [109], may offer more consistent improvements for complex zero-shot clinical reasoning.

Table 6: Performance comparison of different models with Direct Inference vs. Zero-shot CoT.

| Model | Qwen2.5-VL-7B | HuatuoGPT-Vision-7B | HuatuoGPT-Vision-34B | GPT-4o | Gemini-2.5-Pro |
|---|---|---|---|---|---|
| Direct inference | 27.63 | **35.57** | **39.58** | 41.69 | 49.53 |
| Zero-shot CoT | **32.35** | 28.53 | 32.14 | **42.11** | **61.67** |

Second, as highlighted in our findings (Observation 2), domain-specific Supervised Fine-Tuning (SFT) represents a powerful method for instilling specialized knowledge. Future work should prioritize fine-tuning MLLMs on high-quality, curated medical instruction datasets. Such datasets could comprise clinical dialogues, endoscopic procedural reports, and synthetic question-answer pairs tailored to endoscopic analysis. This process is crucial for aligning the model's internal representations and reasoning behavior with the specific nuances and demands of the clinical domain.

Finally, to bolster factual accuracy and mitigate hallucinations, integrating structured medical knowledge is another critical direction. This can be achieved through techniques like Retrieval-Augmented Generation (RAG) [110, 111] or by fusing embeddings from clinical knowledge graphs [112]. By dynamically providing the model with contextual support from verified medical sources during inference, these methods can significantly reduce factual inaccuracies and enhance the reliability of generated outputs.

While a comprehensive implementation of these approaches extends beyond the scope of this benchmark, they collectively outline a clear roadmap for future research. Pursuing these directions is essential for developing MLLMs that are safe, effective, and clinically viable.

# E Potential Negative Social Impacts

We propose a comprehensive benchmark for MLLMs in endoscopy by integrating twenty public endoscopy datasets, which cover diverse endoscopic image types and clinical terminology. Regarding social implications, potential negative impacts may include:

- *Diagnostic Inequality Risks.* The anonymization of public datasets, while protecting patient privacy, results in the loss of critical demographic information (e.g., area, ethnicity). This data gap may lead to biased AI models performing unequally across different population groups, potentially exacerbating healthcare disparities. For instance, the model might achieve higher diagnostic accuracy for specific demographic groups while underperforming for others, particularly underrepresented populations.

- *Technological Exclusion of Underserved Healthcare Systems.* The current benchmark's focus on advanced endoscopic modalities, including wireless capsule endoscopy, risks marginalizing medical institutions in low-resource settings that lack access to such technologies. This creates an infrastructural bias in AI development, where models are optimized primarily for well-equipped hospitals while failing to address the diagnostic needs of resource-constrained clinics. Consequently, the benefits of AI-assisted endoscopy may disproportionately favor high-income regions, exacerbating global healthcare inequities. To promote inclusive progress, future benchmarks should evaluate model performance across diverse clinical environments—from basic to advanced endoscopic systems—ensuring these tools remain accessible and effective regardless of a facility's technological capacity.

- *Security Vulnerabilities in MLLMs Diagnostics.* Current benchmarks exhibit security vulnerabilities by failing to reject harmful inputs, potentially generating dangerous misdiagnoses. These systems lack robust safeguards against adversarial attacks, enabling malicious actors to induce false medical outputs. Such flaws could compromise patient safety through weaponized misinformation or engineered diagnostic errors. Addressing these risks requires specialized medical guardrails and rigorous adversarial testing for clinical deployment.

- *Ethical Implications.* Deploying MLLMs in endoscopic scenarios raises ethical concerns such as patient privacy, model transparency, and accountability, which could lead to misdiagnosis or harm. Thus, developing robust safeguards and clear responsibility frameworks is necessary. Specifically, data collection must have approval from the respective institutional review boards, with newly added data de-identified to remove all patient-related information. For private databases prohibiting direct data use with external APIs like OpenAI, more capable models such as ChatGPT-o1 can be tested via APIs through HIPAA-compliant cloud providers like Azure. The medical benchmark must undergo rigorous clinical review by experienced physicians.

**Correct Sample**

**Question: Assess this endoscopic image for any pathological findings. Should abnormalities be identified, what classifications do they fall under?**

**Options:**
A. Colonoscopy
B. Colon polyp
C. Angiectasia
D. Gastric polyp

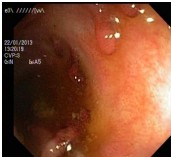

**Gemini-2.5-Pro Response:**

The image shows an endoscopic view of the gastrointestinal tract with raised, irregular, nodular lesions and erythematous mucosa.Options:A. Colon polyp: Abnormal growths projecting from the mucous membrane, common in colonoscopies. The raised, irregular lesions match this appearance.B. Angiectasia: Dilated blood vessels, appearing as bright red, flat or slightly raised lesions with a fern-like pattern. The image lacks this characteristic pattern.C. Gastric polyp: Growths in the stomach lining, similar to colon polyps. Without location context, differentiation is challenging.D. Barrett's esophagus: Esophageal condition with salmon-pink or reddish patches. The image lacks clear esophageal features or typical Barrett's appearance.The raised, irregular lesions suggest polyps. The background folds could indicate colonic haustra, supporting a colon location. Thus, "colon polyp" fits best.

Figure 9: Correct sample

---

**Correct Sample**

**Question: What specific therapeutic procedure is depicted in this image?**

**Options:**
A. Resection margins
B. Resected polyps
C. Dyed resection margins
D. Dyed lifted polyps

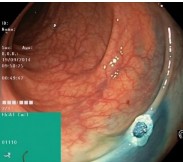

**Gemini-2.5-Pro Response:**

The image shows an endoscopic view, likely of the colon, with a treated circular area in the lower right stained with blue dye.
Options:
A. Resection margins: Edges of tissue removed during resection, plausible for the area around a polyp removal site.
B. Resected polyps: Refers to removed polyps, but the image shows the removal site, not the polyps, so incorrect.
C. Dyed resection margins: Resection margins stained with dye. The blue-stained circular area suggests a resection site with dyed margins, very plausible.
D. Dyed lifted polyps: Involves lifting polyps with a solution before resection. The image shows a post-resection defect, not a lifted polyp, so less likely.Option C is more accurate than A, as the blue dye highlights the resection margins, likely for visualizing complete removal.
Final Answer C

Figure 10: Correct sample

**Correct Sample**

**Question: Which colored rectangle in this endoscopic visualization most accurately indicates the Prograsp Forceps?**

**Options:**
**A. Yellow**
**B. Red**
**C. Green**
**D. Blue**

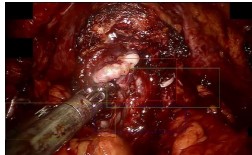

**Gemini-2.5-Pro Response:**

The Prograsp Forceps is a surgical instrument used for grasping, with the tips or jaws being the most characteristic part. The name "Prograsp" emphasizes the grasping action. The rectangle highlighting the jaws best identifies the instrument.
Options:
A. Yellow rectangle: On the tissue, not the forceps.
B. Red rectangle: On the tip/jaws of the Prograsp Forceps, crucial for identifying the grasping function.
C. Green rectangle: On the tissue, not the forceps.
D. Blue rectangle: On the shaft of the Prograsp Forceps, less distinctive than the jaws.
The red rectangle (B) highlighting the jaws is the most accurate for identifying the Prograsp Forceps.
Final Answer B

Figure 11: Correct sample

---

**Correct Sample**

**Question: What is the designation of the gastrointestinal organ depicted in this image?**

**Options:**
**A. Esophagus**
**B. Large intestine**
**C. Stomach**
**D. Small intestine**

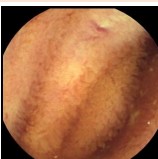

**GPT-4o Response:**

To determine the designation of the gastrointestinal organ depicted in the image, let's analyze the visual characteristics:
1.  Large Intestine: The large intestine typically has a smooth surface with haustra, which are sac-like pouches. The image does not show these features.
2.  Stomach: The stomach lining is characterized by rugae, which are folds that allow the stomach to expand. The image does not show these folds.
3.  Esophagus: The esophagus has a smooth, muscular tube appearance without villi or folds like those seen in the image.
4.  Small Intestine: The small intestine is lined with villi, which are small, finger-like projections that increase the surface area for absorption. The image shows a surface with structures resembling villi.
Based on the presence of villi-like structures, the organ depicted in the image is most likely the small intestine.

Therefore, the correct answer is: D. small intestine.

Figure 12: Correct sample

**Correct Sample**

**Question: Based on the endoscopic visualization, could you assess the quantity of polyps present?**

**Options:**
**A. 0**
**B. 1**
**C. 2**
**D. 3**
**E. 4**
**F. More than 4**

**GPT-4o Response:**

To assess the quantity of polyps present in the provided endoscopic image, let's carefully examine the visual details:
1. Observation: The image shows the interior of a colon, which is the typical setting for identifying polyps during a colonoscopy.
2. Identification of Polyps: Polyps generally appear as small, protruding growths on the lining of the colon. They can vary in size and shape but are often round or oval and can be sessile (flat) or pedunculated (on a stalk).
3. Counting the Polyps: In the image, there are visible protrusions that resemble polyps. We need to count these distinct protrusions to determine the number of polyps.   There is one prominent polyp-like structure visible in the center of the image. There is another smaller polyp-like structure visible to the right of the first one.
4. Conclusion: Based on the visual evidence, there are two distinct polyp-like structures visible in the image.
Therefore, the quantity of polyps present is 2. The correct answer is: C.

Figure 13: Correct sample

**Correct Sample**

Question: Based on the endoscopic image provided, which surgical tool is highlighted within the blue delineated area?

Options:
A. Bipolar Forceps
B. Prograsp Forceps
C. Laparoscopic Grasper
D. Suction Instrument

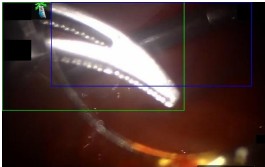

GPT-4o Response:

To identify the surgical tool highlighted within the blue delineated area, let's analyze the image and the options:
1. Bipolar Forceps: Typically have two tips that are used for coagulating tissue. They often have a distinctive shape for grasping and applying electrical current.
2. Prograsp Forceps: Designed for grasping and manipulating tissue. They have a specific design with a strong grip.
3. Laparoscopic Grasper: Used for holding and manipulating tissues during laparoscopic surgery. They have a simple design with a gripping mechanism.
4. Suction Instrument: Used for removing fluids. It usually has a hollow tube design.

In the image, the tool within the blue area appears to have a gripping mechanism with serrated edges, which is characteristic of forceps used for grasping. The design and structure are more consistent with a Prograsp Forceps.
Therefore, the surgical tool highlighted is likely: B. Prograsp Forceps.

Figure 14: Correct sample

**Perceptual Errors**

**Question: What specific abnormality can be discerned from the findings presented in this endoscopic image?**

**Options:**
**A. Erosion**
**B. Polyp**
**C. Ulcer**
**D. Foreign body**

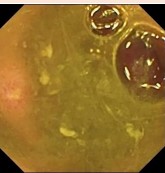

**QvQ Response:**

The image shows a mucosal surface with small, yellowish-white, irregular-shaped particles scattered on it.
Options:
A. Erosion: Superficial mucosal breaks, often with whitish or yellowish exudate. The particles appear particulate and lumpy, not like a typical fibrinous coating.
B. Polyp: Growth protruding from the mucosa. The objects seem to rest on the surface, not attached growths.
C. Ulcer: Deeper mucosal break with a crater-like appearance. The particles lack depth or ulcer characteristics.
D. Foreign body: Extraneous objects like undigested food or pill fragments.
**The yellowish-white, lumpy particles match this description.The particulate objects best fit "foreign body."** The erythematous mucosa may suggest inflammation, but the highlighted abnormality is foreign bodies.
Final Answer D

Figure 15: Error sample

**Perceptual Errors**

**Question: Could you determine the therapeutic procedure depicted in this image?**

**Options:**
A. Dyed lifted polyps
B. Resection margins
C. Resected polyps
D. Dyed resection margins

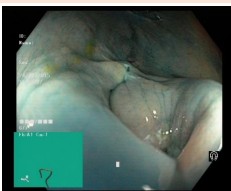

**HuatuoGPT-Vision-34B Response:**

To determine the therapeutic procedure depicted in this image, let's analyze each option step by step:
A.   dyed lifted polyps - This suggests that there are polyps present which have been stained (dyed) and then elevated for removal or examination. The presence of a dye might be indicated if we see distinct coloration on certain areas within the tissue. **However, no such staining is evident here.**
B.   resection margins - Resection margins refer to the edges of the resected area where cancerous cells may still remain after surgery. They would typically not appear as clearly defined structures unless marked with surgical tools or dyes.
C.   resected polyps - If polyps were removed from the mucosal surface, one could expect to see clear cuts or disruptions along their former locations. There doesn't seem to be any obvious signs of recent resection marks like bleeding spots or exposed submucosa layers.
D.   dyed resection indicates that the margins are highlighted using dye. Again, without visible evidence of specific dye application, this seems less likely.
Given the absence of distinctive features associated with options A, C, and D, the most plausible answer based on what can be observed in the provided image appears to align more closely with option B, focusing on examining the potential remaining margins post-resection rather than highlighting them through dye or showing actual resections.

Figure 16: Error sample

## Lack of Knowledge

**Question: Examine this endoscopic image and indicate whether any pathological findings are present. If so, please identify the specific type of abnormality.**

**Options:**
**A. Vascular abnormalities**
**B. Parasitosis**
**C. Ulcer**
**D. Erosion**

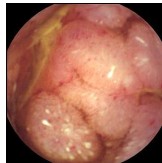

**QvQ Response:**

I've got an endoscopic image of reddish-pink tissue, likely from the digestive tract, with a bumpy, nodular appearance and darker red spots.
The question asks if there are pathological findings and to identify them: vascular abnormalities, parasitosis, ulcer, or erosion.
Options:
A. Vascular abnormalities: Dilated blood vessels or malformations, appearing reddish or purple. **The image shows prominent blood vessels** and dark red spots, suggesting increased vascularity.
B. Parasitosis: Parasitic infections causing inflammation or visible parasites. No obvious parasites or worm-like structures are seen.
C. Ulcer: Breaks in the mucosa with a white or yellow base and red border. No distinct ulcerations are visible.
D. Erosion: Smaller, shallower mucosal breaks, appearing as red, raw areas. Some rough textures could suggest erosions, but less prominent than vascular changes.
**The nodularity and dark red spots suggest vascular abnormalities, possibly indicating bleeding or dilated vessels.** Erosions are possible but less likely as the primary finding.
Final Answer A

Figure 17: Error sample

## Lack of Knowledge

**Question: Does this endoscopic image reveal any pathological findings? If so, could you identify the specific type?**

**Options:**
**A. Angiectasia**
**B. Polyp**
**C. Bleeding**
**D. Erosion**

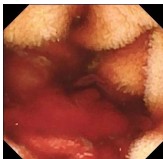

**HuatuoGPT-Vision-34B Response:**

To analyze the endoscopic image and identify any pathological findings, let's consider each option:

A. Angiectasia: Angiectasias are small dilated blood vessels that can appear as red spots or patches on mucosal surfaces in certain conditions. They often have a characteristic appearance of tiny red dots or blotches.

B. Polyp: Polyps are growths protruding from the lining of organs such as the colon, stomach, or nasal passages.

C. Bleeding: Bleeding would be indicated by fresh blood visible within the lumen (interior space) of the organ being examined.

D. Erosion: Erosions refer to superficial breaks or ulcerations in the mucosa which might result from various causes including inflammation, infection, or physical trauma.

Upon examining the provided image: There is no clear evidence of angiectasia due to their typical presentation. No well-defined mass consistent with a polyp is observed. **While there appears to be some reddish discoloration, it does not clearly indicate active bleeding.** However, the disrupted and irregular tissue texture suggests possible erosion. Based on these observations, the most likely pathological finding depicted in this endoscopic image is: D. erosion.

Figure 18: Error sample

**Irrelevant Response**

**Question: Based on the endoscopic visualization, could you ascertain the number of surgical instruments depicted?**

**Options:**
**A. 0**
**B. 1**
**C. 2**
**D. 3**

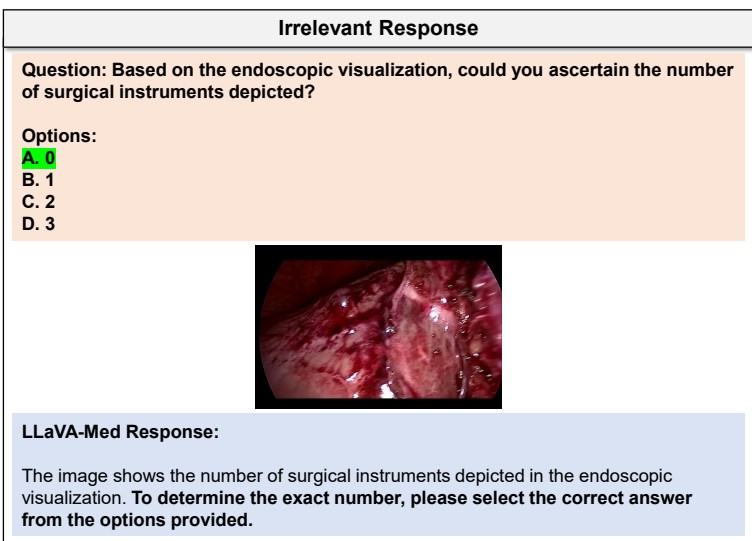

**LLaVA-Med Response:**

The image shows the number of surgical instruments depicted in the endoscopic visualization. **To determine the exact number, please select the correct answer from the options provided.**

Figure 19: Error sample

**Irrelevant Response**

**Question: Analyze this endoscopic image for any pathological findings. Should any be identified, what classifications do they fall under?**

**Options:**
**A. No abnormality**
**B. Bleeding**
**C. Lymphangiectasia**
**D. Erosion**

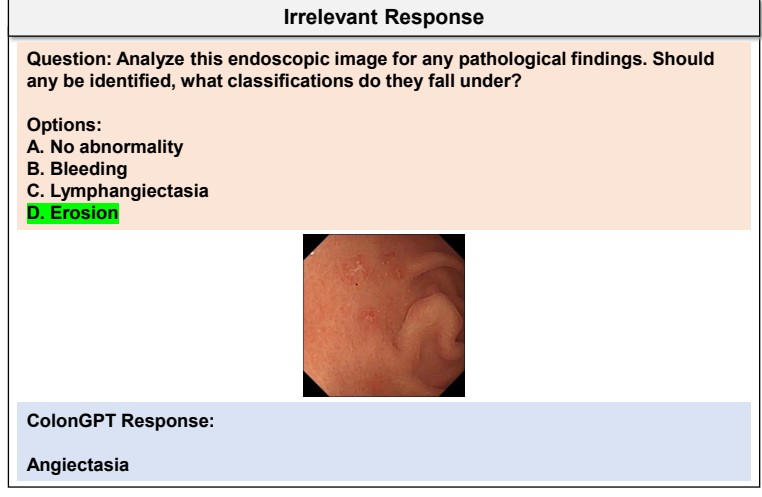

**ColonGPT Response:**

**Angiectasia**

Figure 20: Error sample

**Refusal to Answer**

**Question: Could you identify the coordinates of the low-grade adenoma depicted in this endoscopic image? The required format for the response is [x1, y1, x2, y2].**

**Options:**
**A. [251, 480, 343, 586]**
**B. [529, 392, 617, 488]**
**C. [488, 838, 584, 954]**
**D. [884, 149, 977, 250]**

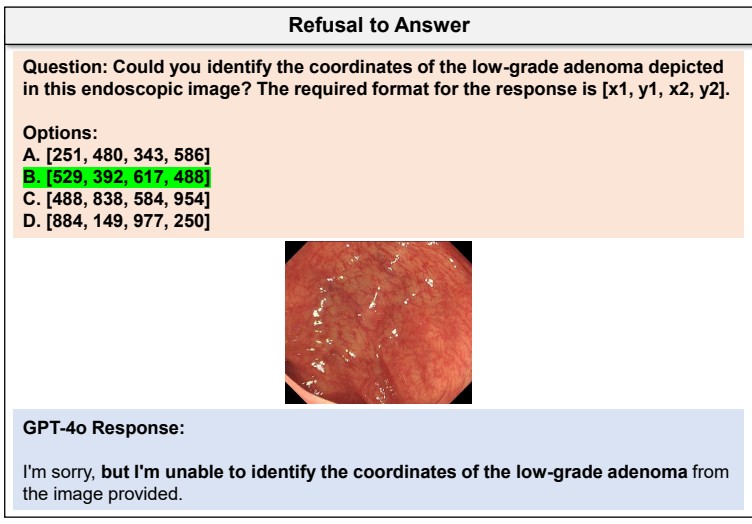

**GPT-4o Response:**

I'm sorry, **but I'm unable to identify the coordinates of the low-grade adenoma** from the image provided.

Figure 21: Error sample

**Refusal to Answer**

**Question: In this endoscopic image, what is the total number of surgical instruments that can be identified?**

**Options:**
**A. 0**
**B. 1**
**C. 2**
**D. 4**

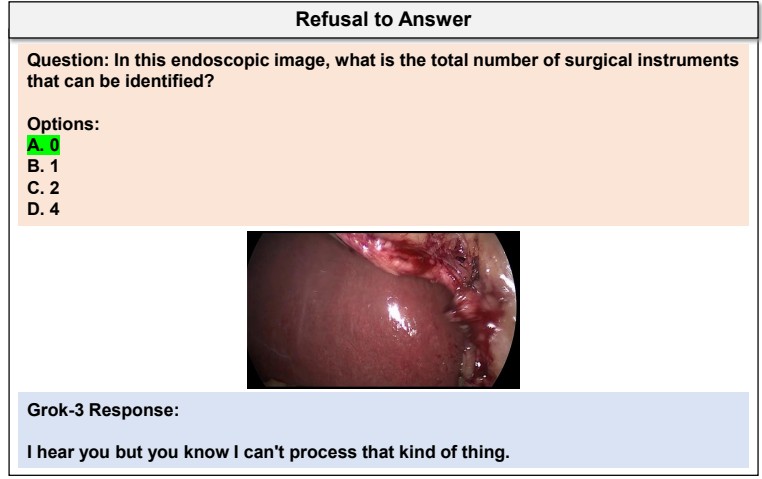

**Grok-3 Response:**

**I hear you but you know I can't process that kind of thing.**

Figure 22: Error sample

# NeurIPS Paper Checklist

The checklist is designed to encourage best practices for responsible machine learning research, addressing issues of reproducibility, transparency, research ethics, and societal impact. Do not remove the checklist: **The papers not including the checklist will be desk rejected.** The checklist should follow the references and follow the (optional) supplemental material. The checklist does NOT count towards the page limit.

Please read the checklist guidelines carefully for information on how to answer these questions. For each question in the checklist:

- You should answer [Yes] , [No] , or [NA] .
- [NA] means either that the question is Not Applicable for that particular paper or the relevant information is Not Available.
- Please provide a short (1–2 sentence) justification right after your answer (even for NA).

**The checklist answers are an integral part of your paper submission.** They are visible to the reviewers, area chairs, senior area chairs, and ethics reviewers. You will be asked to also include it (after eventual revisions) with the final version of your paper, and its final version will be published with the paper.

The reviewers of your paper will be asked to use the checklist as one of the factors in their evaluation. While "[Yes] " is generally preferable to "[No] ", it is perfectly acceptable to answer "[No] " provided a proper justification is given (e.g., "error bars are not reported because it would be too computationally expensive" or "we were unable to find the license for the dataset we used"). In general, answering "[No] " or "[NA] " is not grounds for rejection. While the questions are phrased in a binary way, we acknowledge that the true answer is often more nuanced, so please just use your best judgment and write a justification to elaborate. All supporting evidence can appear either in the main paper or the supplemental material, provided in Appendix. If you answer [Yes] to a question, in the justification please point to the section(s) where related material for the question can be found.

IMPORTANT, please:

- **Delete this instruction block, but keep the section heading "NeurIPS Paper Checklist",**
- **Keep the checklist subsection headings, questions/answers and guidelines below.**
- **Do not modify the questions and only use the provided macros for your answers**.

