# OpenReview forum: "EndoBench: A Comprehensive Evaluation of Multi-Modal Large Language Models for Endoscopy Analysis"
_NeurIPS.cc/2025/Datasets_and_Benchmarks_Track — NeurIPS 2025 Datasets and Benchmarks Track poster_

### Official Review · Reviewer_BaSf · 2025-06-16

**Rating:** 5
**Confidence:** 4

**Summary:**

This work proposes EndoBench, a multimodal benchmark designed for large foundation models in endoscopic scenarios, covering four major tasks and multiple data modalities. The paper compiles and annotates two large-scale public endoscopy video datasets containing over 7,000 videos and a total of 8 million frames, with standardized preprocessing. The authors further evaluate 23 MLLMs on this benchmark, revealing notable performance disparities and challenges in endoscopic tasks.

**Additional Feedback:**

None.

**Dataset Code Accessibility:**

Yes

**Dataset Code Comments:**

I can run them.

**Ethical Comments:**

All included data are from existing publicly available datasets, which were only standardized and re-annotated. The original data have been de-identified, complying with ethical requirements.

**Ethical Considerations:**

No, there are no or only very minor ethics concerns

**Final Justification:**

I have been optimistic about this work from the beginning, as the dataset, code, and clinical relevance all merit publication. The authors have provided detailed clarification and additions regarding dataset details and subgroup analysis, and have added a case study section as I suggested. Therefore, I maintain my score of 5. I hope this study will contribute to further advancement in the related field.

**Limitations Weaknesses:**

1. Lack of discussion on clinical relevance. Although the dataset covers multiple tasks, some of them (e.g., the design of VQA questions) lack a detailed explanation of their practical significance and application scenarios in clinical workflows. It is recommended to add more connections to real surgical contexts, such as surgical navigation or anomaly detection support.

2. The paper would benefit from additional error case analysis to further uncover the structural challenges faced by current MLLMs.

3. The distribution of different types of surgeries or anatomical regions in the dataset is not clearly described. This may result in model overfitting to certain areas and underfitting to others. Including more detailed statistics on data distribution is recommended.

4. There is a lack of systematic comparison with existing multimodal medical benchmarks.

**Strengths Contributions:**

1. The task and modality design is comprehensive. EndoBench covers representative tasks across the diagnostic and therapeutic workflow, including multiple data modalities (such as 3D pose and depth), which enables a thorough evaluation of multimodal large models in real clinical videos.

2. The dataset is large and processed in a standardized manner.

3. A unified evaluation protocol and baseline model reports are provided, facilitating future comparisons and evaluations.

4. The manuscript is well-structured, with rich and intuitive figures and tables.

5. It covers multiple downstream subtasks: beyond vision-language matching and question answering, it also includes image analysis (detection/segmentation) and navigation (pose estimation), expanding the evaluation scope for medical foundation models.

---

> ### Author Rebuttal · Authors · 2025-07-29
>
> We sincerely thank the reviewer for recognizing EndoBench’s significance, comprehensive design, and clear presentation,  and we will respond to each concern point-by-point:
>
> ## Q1
> >Lack of discussion on clinical relevance.
>
> Thank you for the valuable feedback. To demonstrate the clinical relevance of EndoBench tasks, we elaborate on how all 12 tasks support a complete standard endoscopic workflow in typical clinical practice. The procedure follows three critical phases: 1) **diagnostic assessment**, 2) **surgical planning**, and 3) **therapeutic implementation**, with each EndoBench task directly supporting specific clinical needs throughout this progression.
>
> 1) During the **diagnostic assessment phase**, when a patient presents with gastrointestinal symptoms requiring endoscopic evaluation, the procedure begins with a systematic examination. **Organ Identification** provides real-time anatomical orientation as the endoscope navigates between esophagus, stomach, and duodenum, while **Landmark Identification** recognizes critical structures like pylorus or ampulla of Vater to ensure complete coverage. Upon detecting abnormalities, **Lesion Type Identification** characterizes pathology (polyps, ulcers, inflammation), **Lesion Quantification** counts multiple lesions for treatment planning, and **Lesion Severity Grading** applies standardized scoring systems (Mayo Score, BBPS) to guide therapeutic decisions.
>
> 2) The workflow progresses to the **surgical planning phase**, where **Visual Grounding** provides precise coordinates for surgical navigation systems, **Region Selection** identifies accurate target localization, **Region Recognition** characterizes specific structures within regions of interest, and **Preoperative Assessment** evaluates procedural safety parameters before intervention.
>
> 3) During the **therapeutic implementation phase**, **Macro Phases Identification** provides real-time workflow guidance across surgical stages (dissection, resection, suturing), **Instrument Management** ensures safety by monitoring instrument presence and preventing retained equipment, and *Micro Operation Analysis** analyzes specific instrument functions (grasping, cutting, cauterization) for objective skill assessment and quality assurance.
>
> This complete workflow integration ensures that MLLMs are evaluated on capabilities that directly translate to improved patient care, surgical navigation support, and clinical decision-making in real-world endoscopic practice. We will incorporate this end-to-end workflow mapping into the revised version to better highlight the clinical value and practical relevance of each task in EndoBench.
>
> ## Q2
> >The need for an additional error case analysis.
>
> Thank you for your valuable suggestion. We fully agree that a deeper analysis of model failure cases is crucial for uncovering the structural challenges faced by current MLLMs in medical applications. In response, we will add a dedicated **Case study Section** in the revised version to provide a more granular understanding of typical error patterns and their underlying causes.
>
> The observed errors in our case studies are categorized into **four representative types**, each highlighting a specific limitation of current MLLMs in clinical endoscopy settings. Due to rebuttal policy constraints, we are unable to include visual samples at this stage. However, we will introduce these error types in the revised manuscript and provide detailed text-image examples and discussion to enhance interpretability.
>
> 1) **Perceptual Errors**: These arise when MLLMs fail to correctly perceive or interpret critical visual elements, such as missing a lesion, misidentifying anatomical structures, or overlooking subtle yet important patterns. This reveals weaknesses in the model’s low-level visual comprehension.
>
> 2) **Lack of Knowledge**: In some cases, models successfully identify visual content and understand the question but still generate incorrect answers due to insufficient domain-specific medical knowledge. This includes misinterpreting clinical signs or confusing visually similar conditions.
>
> 3) **Irrelevant Response**: The model may produce answers that are off-topic, vague, or incoherent, indicating a disconnect between the visual input, the question, and the generated response. These failures underscore the need for tighter alignment between perception, reasoning, and language output.
>
> 4) **Refusal to Answer**: Certain models—especially proprietary ones—intentionally decline to respond when faced with questions involving sensitive topics, ambiguous instructions, or content deemed outside the model’s safe response boundaries. While this reflects safety-oriented design, it also limits clinical usability in nuanced scenarios.
>
> We believe that adding this structured error analysis will provide valuable insights into model limitations and offer concrete guidance for future improvements in MLLM development for medical applications.
>
> ## Q3
> >The distribution of different types of surgeries or anatomical regions in the dataset is not clearly described.
>
> Thank you for your insightful suggestion. We fully agree that a clear and detailed description of the dataset distribution is essential to ensure fair and robust benchmarking. To address this, our benchmark included detailed statistics across four distinct endoscopy scenarios—**gastroscopy, colonoscopy, capsule endoscopy, and surgical endoscopy**—which we selected to reflect the diverse visual domains encountered in clinical practice. **These scenarios exhibit notable domain shifts in image characteristics, tools, and visual context, making them an important axis of variation in our evaluation.**
>
> In response to your comment, we further analyzed the Organ Identification task and found that the dataset achieves a relatively balanced distribution across key anatomical regions: **small intestine (26.36%), large intestine (23.10%), stomach (25.54%), and esophagus (25.00%)**. This balanced composition helps ensure that model evaluation is not biased toward any single anatomical structure.
>
> Regarding surgical procedures, the surgical endoscopy subset includes four representative surgery types:
>
> - Abdominal porcine surgeries (30.77%) from EndoVis17 and EndoVis18,
> - Human laparoscopic cholecystectomy (16.05%) from Cholec80,
> - Human radical prostatectomy (41.98%) from PSI-AVA, and
> - Endoscopic mucosal resection (11.20%) from Kvasir, HyperKvasir, and GastroVision.
>
> We will incorporate these detailed statistics, along with supporting tables and visualizations, in the revised version of the paper to provide greater transparency into the dataset structure. This addition will also help readers interpret model performance differences with greater nuance and identify potential dataset biases. Once again, we appreciate your valuable feedback in helping improve the clarity and completeness of our benchmark.
>
> ## Q4
> >There is a lack of systematic comparison with existing multimodal medical benchmarks.
>
> Thank you for your valuable advice. We agree that a systematic comparison with existing multimodal medical benchmarks is essential to highlight the unique contributions of our work.
>
> **In Table 1**, we provide a comprehensive comparison with existing multimodal endoscopic benchmarks, emphasizing the broader coverage of tasks, clinical scenarios, and visual prompting granularities in EndoBench. Furthermore, in **Section 2.2 (line 116–128)**, we discuss existing multimodal medical benchmarks, which are primarily focused on imaging modalities such as CT, X-ray, and pathology. Compared to these modalities, endoscopic images are significantly underrepresented; existing benchmarks in this domain are typically limited to narrow tasks such as instrument detection during surgery or abnormality detection (e.g., polyps or lesions), often with small-scale datasets and restricted to a single procedural context. In contrast, EndoBench offers a more holistic evaluation framework that spans a wide spectrum of endoscopic applications.
>
> We will further emphasize this contrast in the revised version and elaborate on how EndoBench complements and extends the scope of existing benchmarks, providing a much-needed resource to evaluate and advance MLLMs in real-world endoscopic workflows.

---

> > ### Comment · Reviewer_BaSf · 2025-08-02
> >
> > Thanks for the thorough response. I will maintain my final score.

---

> > > ### Author Response · Authors · 2025-08-06
> > > **Official Comment by Authors**
> > >
> > > We sincerely appreciate your time and effort in reviewing our paper.

---

### Official Review · Reviewer_yyLh · 2025-07-01

**Rating:** 4
**Confidence:** 3

**Summary:**

This paper presents EndoBench, a comprehensive benchmark for evaluating multi-modal large language models (MLLMs) in endoscopic image analysis. EndoBench covers four endoscopic scenarios, 12 specialized clinical tasks, and five levels of visual prompting granularities, resulting in 6,832 clinically validated VQA pairs from 21 diverse datasets. The benchmark aims to holistically evaluate the perceptual and diagnostic abilities of MLLMs in realistic clinical workflows. The authors benchmark 23 state-of-the-art models, including general-purpose, medical-specialized, and proprietary MLLMs, and establish human clinician performance as a reference standard. The results reveal that proprietary MLLMs outperform open-source and medical-specialized models but still trail human experts, highlighting the need for further advancements in domain adaptation and prompt design.

**Dataset Code Accessibility:**

Partly

**Ethical Considerations:**

No, there are no or only very minor ethics concerns

**Limitations Weaknesses:**

1. Despite the advancements, all models still significantly underperform compared to human clinicians, especially in complex clinical scenarios. This underscores the limitations of current MLLMs in clinical diagnosis and spatial understanding.
2. Some medical domain-specific models show limitations in instruction-following capabilities and overfitting, restricting their performance in broader application scenarios.
3. The benchmark reveals severe limitations in polyp counting tasks, with no model achieving above 30% accuracy. This highlights the dual challenges of lesion identification and numerical reasoning in current MLLMs.

**Strengths Contributions:**

1. EndoBench is the most extensive multi-modal endoscopic benchmark to date, covering four distinct endoscopic scenarios and 12 specialized clinical tasks. This comprehensive coverage allows for a more accurate assessment of MLLMs' clinical utility.
2. The dataset consists of 6,832 clinically validated VQA pairs, ensuring the relevance and accuracy of the evaluation. This rigorous validation process enhances the credibility of the benchmark.
3. The extensive experiments provide valuable insights into the current capabilities and limitations of MLLMs in endoscopic analysis. The findings highlight the gap between current models and human expertise, guiding future research directions.

---

> ### Author Rebuttal · Authors · 2025-07-29
>
> Thank you for your positive evaluation of EndoBench, particularly your recognition of our comprehensive benchmarking and clinically aligned task design. Below, we address the remaining concerns you have raised:
>
> >1. Despite the advancements, all models still significantly underperform compared to human clinicians, especially in complex clinical scenarios. This underscores the limitations of current MLLMs in clinical diagnosis and spatial understanding.
>
> **Response**: We thank the reviewer for this observation and completely agree with this assessment. Indeed, the significant underperformance of current MLLMs compared to human clinicians is precisely what our comprehensive evaluation reveals and represents one of our key findings. As we discuss in our paper **(line 243-252 in the Discussion section)**, this performance gap underscores exactly why EndoBench is needed, to provide a rigorous, standardized evaluation that can guide future research toward bridging these clinical gaps. We appreciate the reviewer highlighting this important limitation of current technology that our benchmark successfully exposes.
>
>
> >2. Some medical domain-specific models show limitations in instruction-following capabilities and overfitting, restricting their performance in broader application scenarios.
>
> **Response**: We thank the reviewer for highlighting this important observation. We completely agree with this assessment regarding medical domain-specific models. As we discuss in our analysis **(line 253-261)**, medical models that underwent domain-specific supervised fine-tuning, while showing exceptional performance in certain tasks like landmark identification and organ recognition, do indeed exhibit limitations in instruction-following capabilities and suffer from overfitting issues. This finding emphasizes the need for more balanced approaches in medical MLLM development that preserve both domain expertise and general instruction-following abilities. Our benchmark's multi-dimensional capability evaluation framework is designed to capture exactly these types of trade-offs, helping guide future research toward more robust medical AI systems.
>
> >3. The benchmark reveals severe limitations in polyp counting tasks, with no model achieving above 30% accuracy. This highlights the dual challenges of lesion identification and numerical reasoning in current MLLMs.
>
> **Response**:  We thank the reviewer for emphasizing this critical finding. We completely agree that the severe limitations in polyp counting (no model above 30% accuracy) represent a significant challenge for current MLLMs. As we analyze in our discussion **(line 272-280)**, this task exposes dual challenges in both lesion identification and numerical reasoning. Our investigation revealed that when we provided bounding boxes as visual prompts, accuracy dramatically improved across all models (e.g., Gemini-2.5-Pro from 24.46% to 92.57%), demonstrating that the primary bottleneck lies in lesion identification rather than counting ability. This finding highlights the critical need for enhanced domain-specific medical knowledge in MLLMs and underscores the value of our benchmark in identifying such specific capability gaps that can guide targeted improvements in future model development.

---

> > ### Author Response · Authors · 2025-08-06
> > **Follow-up on Previous Response**
> >
> > Dear Reviewer,
> >
> > Hope this message finds you well. As the discussion period is nearing its end with less than three days remaining, I wanted to ensure we have addressed all your concerns satisfactorily. If you have any additional points or feedback you'd like us to consider, please let us know. Your insights are invaluable to us, and we are eager to address any remaining issues to improve our work.
> >
> > Thank you for your time and effort in reviewing our paper.

---

### Official Review · Reviewer_NhPZ · 2025-07-01

**Rating:** 5
**Confidence:** 4

**Summary:**

The paper introduces EndoBench, a comprehensive benchmark for evaluating multi-modal large language models (MLLMs) across diverse endoscopic procedures, including gastroscopy, colonoscopy, capsule endoscopy, and surgical endoscopy. EndoBench includes 6,832 clinically validated VQA pairs from 21 diverse datasets, covering a broad range of tasks such as anatomical recognition, lesion analysis, spatial localization, and surgical operations. The paper also benchmarks 23 state-of-the-art models, including general-purpose, medical-specialized, and proprietary models. The authors find that while proprietary models outperform open-source and medical-specialized models, all models still trail human experts in performance.

**Dataset Code Accessibility:**

Partly

**Ethical Considerations:**

No, there are no or only very minor ethics concerns

**Final Justification:**

The response solved my concerns, and the paper was well written. I am willing to increase my score.

**Limitations Weaknesses:**

1.Limited Novelty in Methodology:
While the benchmark itself is a valuable resource, the paper lacks novel methodologies in terms of model improvement or new techniques to address current limitations. The paper mainly focuses on evaluating existing models without providing new solutions for addressing the challenges identified, some training-free methods or insights can be included.

2.Insufficient Focus on Model Improvements:
The paper identifies that proprietary models outperform others but does not propose substantial methods for mitigating the performance gap between MLLMs and human experts. The paper could be more impactful by suggesting techniques to improve model performance, such as domain adaptation, prompt engineering, or integrating more medical knowledge into the models.

3.Overemphasis on Benchmarking:
While benchmarking is crucial, the paper places too much emphasis on it without providing deeper insights into the reasons behind the models' shortcomings. Understanding the underlying causes of performance gaps, such as limitations in spatial localization or lesion recognition, could make the paper more informative.

4.Lack of Detailed Discussion on Ethical Implications:
The paper does not address the ethical concerns of deploying these models in real-world medical scenarios. As AI continues to play a larger role in healthcare, it is essential to consider ethical issues such as patient privacy, model transparency, and accountability, especially when using these models for diagnosis and surgical procedures.

**Strengths Contributions:**

1.Thorough Evaluation:
The paper presents a detailed evaluation of 23 models, which includes both open-source and proprietary models, allowing for a robust comparison across different types of MLLMs. The authors also provide a comparison to human performance, setting a realistic benchmark for model performance in clinical settings.

2.Clinical Relevance:
The tasks chosen for evaluation, such as lesion quantification and preoperative assessment, align with real clinical workflows, making the benchmark highly relevant for the development of AI in clinical settings.

---

> ### Author Rebuttal · Authors · 2025-07-29
>
> Thank you for the valuable comments and encouraging assessment of EndoBench’s scope and clinical relevance. Below, we address the remaining concerns you have raised:
> ## Q1&Q2
> > Limited Novelty in Methodology and Insufficient Focus on Model Improvements
>
> For Q1 and Q2, we fully agree that methodological innovations are essential for advancing the field. However, the primary goal of our work is to systematically benchmark the performance gaps of existing MLLMs in endoscopic tasks, thereby identifying key limitations and motivating future methodological developments.
>
> In response to your suggestion regarding training-free prompting strategies, we have extended our evaluation to include a comparison between direct inference and Zero-Shot Chain-of-Thought (CoT [1]) prompting on several representative models, including general-purpose, medical-specialized, and proprietary MLLMs. To ensure fair evaluation, we employed LLM-as-a-Judge, with Qwen2.5-VL-72B serving as the judgment model. Our findings reveal that while Qwen-2.5-7B, GPT-4o, and Gemini-2.5-Pro show improved accuracy with CoT prompting, HuatuoGPT experiences a decline in accuracy, indicating that CoT prompting can sometimes harm performance by increasing the risk of hallucinations. Similar observations have been reported in recent literature [2,3,4], suggesting that CoT prompting may not be universally beneficial, especially in tasks requiring precise clinical reasoning.
>
> | Model             | Qwen2.5-VL-7B | HuatuoGPT-Vision-7B | HuatuoGPT-Vision-34B | GPT-4o | Gemini-2.5-Pro |
> |-----------------|:-----------:|:-------------------:|:--------------------:|:------:|:--------------:|
> | Direct inference  |    27.63    |        **35.57**        |        **39.58**         |  41.69 |     49.53      |
> | Zero-shot CoT     |    **32.35**    |        28.53        |         32.14           |  **42.11** |       **61.67**        |
>
> Moreover, for suggesting techniques to improve model performance, we acknowledge the potential of prompt engineering methods such as Self-Consistency (SC) [5], Self-Refine [6], and MedPrompts [7], which enhance model performance by increasing sampling diversity and improving response consistency. These techniques are particularly promising for improving zero-shot reasoning in complex clinical scenarios and can be incorporated without additional model training. In addition, as **highlighted in our paper (Observation 2)**, domain-specific supervised fine-tuning significantly boosts model performance. Future work can further explore this by fine-tuning MLLMs on medical instruction datasets, including clinical dialogues, procedural reports, and synthetic QA pairs tailored to endoscopic tasks, to better align model behavior with clinical reasoning. Another promising direction is the integration of structured medical knowledge through retrieval-augmented generation (RAG) or embedding fusion to provide contextual support during inference [8,9,10]. While these approaches are beyond the current scope of our benchmark, we agree they represent important directions for future research, and we plan to incorporate a more detailed discussion of these strategies in the revised version.
>
> Reference:
>
> [1] Wei, Jason, et al. Chain-of-thought prompting elicits reasoning in large language models. NeurIPS 2022.
>
> [2] Liu, Ryan, et al. Mind your step (by step): Chain-of-thought can reduce performance on tasks where thinking makes humans worse. ICML25.
>
> [3] Sprague, Zayne, et al. To cot or not to cot? chain-of-thought helps mainly on math and symbolic reasoning. ICLR25.
>
> [4] Le, Chenqian, et al. Instruction tuning and cot prompting for contextual medical qa with llms. arXiv preprint arXiv:2506.12182 (2025).
>
> [5] Wang, Xuezhi, et al. Self-consistency improves chain of thought reasoning in language models. arXiv preprint arXiv:2203.11171 (2022).
>
> [6] Madaan, Aman, et al. Self-refine: Iterative refinement with self-feedback. NeurIPS23.
>
> [7] Nori, Harsha, et al. Can generalist foundation models outcompete special-purpose tuning? case study in medicine. arXiv preprint arXiv:2311.16452 (2023).
>
> [8] Wu, Junde, et al. Medical graph rag: Towards safe medical large language model via graph retrieval-augmented generation. arXiv preprint arXiv:2408.04187 (2024).
>
> [9] Xia, Peng, et al. MMed-RAG: Versatile Multimodal RAG System for Medical Vision Language Models. ICLR 2025.
>
> [10] Lopez, Ivan, et al. Clinical entity augmented retrieval for clinical information extraction. npj Digital Medicine 8.1 (2025): 45.
>
> ## Q3
> > Overemphasis on benchmarking without providing deeper insights into the reasons behind the models' shortcomings
>
> Thanks for your constructive feedback. We fully agree that understanding the underlying causes of model shortcomings is essential for deriving actionable insights and guiding future improvements beyond benchmarking alone. Our paper goes beyond surface-level evaluation through targeted analyses presented in **Observation 3 (line 262-271)** and **Observation 4 (line 272-280)**.
>
> In Observation 3, we explore how model performance varies across different visual prompt formats, revealing a critical gap between spatial perception and medical comprehension. While models—especially proprietary ones—perform well in tasks like ROI selection, they struggle with lesion classification within those regions, indicating that limited medical knowledge, rather than visual processing, is the primary bottleneck.
>
> Observation 4 further supports this insight by examining polyp counting tasks, where the poor performance across most models (<30%) is attributed not to spatial reasoning, but to difficulty in lesion identification. Interestingly, prompting improvements significantly boost performance for some models (e.g., Gemini-2.5-Pro reaching 92%), again emphasizing the importance of domain-specific understanding.
>
> To deepen our analysis in the revised version, we plan to incorporate a detailed categorization of model failure types across tasks, including **lack of medical knowledge, refusal to respond, perceptual errors, and irrelevant or hallucinated outputs (Detailed in Reviewer BaSf Q2)**. By explicitly classifying these error types, we aim to offer clearer insights into the specific capabilities and limitations of current MLLMs, and to better inform future directions for model development and adaptation in clinical contexts.
>
> ## Q4
> > Lack of Detailed Discussion on Ethical Implications
>
> We thank you for the constructive comment that helped us to significantly improve our manuscript. We will add the description of the detailed discussion on ethical implications in **Section D of Potential Negative Social Impacts (in Appendix)** in the revised manuscript as follows:
>
> *Ethical Implications.* Deploying MLLMs in endoscopic scenarios raises ethical concerns such as patient privacy, model transparency, and accountability, which could lead to misdiagnosis or harm. Thus, developing robust safeguards and clear responsibility frameworks is necessary. Specifically, data collection must have approval from the respective institutional review boards, with newly added data de-identified to remove all patient-related information. For private databases prohibiting direct data use with external APIs like OpenAI, more capable models such as ChatGPT-o1 can be tested via APIs through HIPAA-compliant cloud providers like Azure. The medical benchmark must undergo rigorous clinical review by experienced physicians.

---

> > ### Comment · Reviewer_NhPZ · 2025-08-05
> >
> > Thank you for your response. I am pleased to see that my concerns have been addressed. I am willing to increase my score accordingly.

---

> > > ### Author Response · Authors · 2025-08-06
> > > **Official Comment by Authors**
> > >
> > > Thank you very much for raising the score. We appreciate your time and effort in reviewing our paper.

---

### Official Review · Reviewer_i3wX · 2025-07-02

**Ethics Flags:** Improper research involving human sub…
**Rating:** 4
**Confidence:** 2

**Summary:**

This paper introduces EndoBench, a comprehensive benchmark for evaluating multi-modal large language models (MLLMs) in endoscopic image analysis. It assesses model performance across tasks like endoscopy, lesion identification, etc. Results show proprietary and domain-adapted MLLMs outperform open-source ones in many tasks but still lag behind human clinicians, especially in complex clinical scenarios. EndoBench reveals limitations in MLLMs’ clinical diagnosis and spatial understanding, highlighting the need for research in domain adaptation and prompt design, aiming to advance clinically relevant AI in endoscopy.

**Additional Feedback:**

NA

**Dataset Code Accessibility:**

Yes

**Ethical Considerations:**

No, there are no or only very minor ethics concerns

**Final Justification:**

4

**Limitations Weaknesses:**

The paper focuses on benchmarking MLLMs in standard endoscopic tasks but lacks exploration of rare or complex clinical variants (e.g., endoscopies in patients with severe comorbidities, atypical anatomical structures). For example, while it tests lesion identification in typical scenarios (Sections on Polyp Counting, Lesion Classification), there’s no analysis of how models perform in cases where lesions overlap with surgical scars or distorted organs (no references to such edge cases in data or results). To address this, the authors could supplement EndoBench with a “challenging cases” subset featuring rare pathologies, anatomical anomalies, or post - surgical endoscopic views.

**Strengths Contributions:**

**1.Significance and Novelty**

EndoBench fills a crucial gap as the most comprehensive benchmark for MLLMs in endoscopic image analysis to date. Prior to this, there was a lack of a unified, extensive evaluation framework for this specific medical domain, and it provides a standardized way to measure model performance.

**2.Potential Impact and Relevance**

It has high potential to drive progress in clinically relevant AI for endoscopy. By exposing model limitations in clinical diagnosis and spatial understanding, it guides future research in domain adaptation (e.g., enhancing medical knowledge infusion) and prompt design, which can directly improve AI-assisted endoscopic diagnosis, potentially benefiting patient care.

---

> ### Author Rebuttal · Authors · 2025-07-28
>
> **Question:**
> >The paper focuses on benchmarking MLLMs in standard endoscopic tasks but lacks exploration of rare or complex clinical variants (e.g., endoscopies in patients with severe comorbidities, atypical anatomical structures). For example, while it tests lesion identification in typical scenarios (Sections on Polyp Counting, Lesion Classification), there’s no analysis of how models perform in cases where lesions overlap with surgical scars or distorted organs (no references to such edge cases in data or results). To address this, the authors could supplement EndoBench with a “challenging cases” subset featuring rare pathologies, anatomical anomalies, or post-surgical endoscopic views.
>
> **Response:**
> We sincerely thank the reviewer for acknowledging the significance, novelty, and potential clinical impact of our work on EndoBench. For the limitation mentioned by the reviewer, we fully agree that evaluating model performance on rare and complex clinical scenarios is critical for assessing robustness and real-world applicability. Edge cases—such as patients with severe comorbidities, anatomical distortions, or post-surgical changes—pose significant challenges even for experienced clinicians and represent valuable test cases for advanced MLLMs.
>
> In response to this suggestion, we are currently developing an extension of our benchmark, which we refer to as **EndoBench-Extend**. This supplementary subset will focus specifically on **challenging and underrepresented cases**, including **rare pathologies, overlapping lesions, atypical anatomical presentations, and post-surgical endoscopic views**. We are curating this subset manually from diverse datasets, such as Kvasir, HyperKvasir, GastroVision, WCE2025, and EndoVis18, with careful selection by medical experts to ensure clinical relevance and diagnostic difficulty. Based on feasibility and the need for expert-level annotation, we expect EndoBench-Extend to contain around **50–100 high-quality open-set question sets**, aiming to simulate real-world diagnostic ambiguity and capture failure modes that standard benchmarks may overlook.
>
> Due to rebuttal policy constraints, we are unable to share the dataset or visual samples at this stage. However, we are committed to releasing EndoBench-Extend publicly in future work to complement the current benchmark and further push the frontier of clinically grounded evaluation. We believe this addition will significantly enhance the comprehensiveness and clinical realism of EndoBench.
>
> The question templates in EndoBench-Extend will primarily fall into two categories: one for **endoscopic diagnostic scenarios**, designed to evaluate the model’s ability to identify anatomical structures and lesions; and another for **surgical scenarios**, aimed at assessing the model’s understanding of surgical procedures and tissue conditions. To illustrate the scope of this extension, we will include cases such as:
>
> 1) **Anatomical anomalies in gastrointestinal endoscopy**: Images of the Ampulla of Vater, a physiological elevation that may be mistaken for pathological lesions like inflammation, particularly when abnormalities are present nearby, making such cases rare and challenging.
>
> 2) **Rare pathological presentations**: Images of Grade D esophagitis, which may involve complex mucosal conditions such as scarring or strictures, rendering diagnosis difficult.
>
> 3) **Deformed organ structures with concurrent pathologies**: Cases like diverticula with inflammation or intestinal strictures with ulcers, where organ deformation complicates lesion identification.
>
> 4) **Surgical scenarios with procedural challenges**: Images from polypectomy or laparoscopic cholecystectomy where bleeding occurs or surgical instruments overlap with tissues. Additionally, distorted anatomical structures in laparoscopic surgery, such as stretched or adherent perirenal fascia, pose significant difficulties.
>
> **Due to rebuttal policy constraints, we are unable to share the dataset or visual samples at this stage.**. However, we are committed to releasing EndoBench-Extend publicly in future work to complement the current benchmark and further push the frontier of clinically grounded evaluation. We believe this addition will significantly enhance the comprehensiveness and clinical realism of EndoBench.

---

> > ### Comment · Reviewer_i3wX · 2025-08-05
> > **ok, good luck✋✋**
> >
> > Authors have solved my concerns, and i will maintain my score.

---

> > > ### Author Response · Authors · 2025-08-06
> > > **Thank you**
> > >
> > > We sincerely appreciate the time and effort you have taken to review our paper.

---

### Note · Authors · 2025-08-14

Dear Area Chair,

We sincerely appreciate the time and effort you took to coordinate our submission. We thank all reviewers for recognizing the key strengths of our work, including:

- **First comprehensive endoscopic benchmark** with a **large-scale, clinically validated dataset** that fills a crucial gap in MLLMs evaluation [reviewers i3wX, yyLh, BaSf].
- **High clinical relevance** with tasks aligned to real clinical workflows. [reviewers NhPZ]
- **Extensive and thorough evaluation** of **23 state-of-the-art models**. [reviewers yyLh, NhPZ]
- **Well-structured manuscript** with **valuable insights** into current capabilities and limitations, guiding future research directions. [reviewers BaSf, yyLh]

During the discussion period, we have addressed all reviewer concerns thoroughly:

**Enhanced Dataset Content:**
- **Challenging subset evaluation:** Added a new subset, the EndoBench-Extend with 50–100 curated cases of rare pathologies and post-surgical views to enhance clinical realism. [Reviewer i3wX]
- **Dataset distribution analysis:** Included statistics on organ and surgery type distribution for transparency. [Reviewer BaSf]

**Additional Experiments:**
- **Error case study:** Added case study on four main error types, including perceptual errors, lack of knowledge, irrelevant response, and refusal to answer. [Reviewers NhPZ, BaSf]
- **CoT prompting experiments:** Compared with direct inference, CoT shows accuracy gains for some models but risks of hallucinations in others. [Reviewer NhPZ]

**Further Discussions:**
- **Ethical Implications:** Expanded on patient privacy, transparency, and accountability with safeguards like data de-identification and HIPAA-compliant testing. [Reviewer NhPZ]
- **Model improvement methods:** Discussed prompt engineering (e.g., Self-Consistency) and fine-tuning strategies like RAG to boost performance. [Reviewer NhPZ]

We commit to incorporating all reviewer feedback into our revised manuscript. Through this rebuttal phase, all reviewers expressed satisfaction with the responses, with one reviewer explicitly mentioning increasing their score to acceptance.

Thank you again for your valuable feedback.

Sincerely,

Authors

---

### Decision · Program_Chairs · 2025-09-18

**Decision:**

Accept (poster)

**Comment:**

This paper introduces a benchmark for evaluating multi-modal LLMs in endocscopy spanning 21 datasets, twelve clinical tasks, different scenarios, and five levels of visual prompting. The strengths of the paper lie in the novelty and clinical relevance, physician validation, and real work flows. The reviewers pointed out weaknesses in novel methodology and reliance on template and LLM generated language for the QA task constructions and the absence of rare or challenging clinical tasks, thoug the authors  have committed to add these. The authors improved the paper in the rebuttal by adding new analyses, statistics, and experiments. Overall, I think this benchmark is of high value to the medical AI community and recommend accept.